# VariFace: Fair and Diverse Synthetic Dataset Generation for Face Recognition

## Abstract

The use of large-scale, web-scraped datasets to train face recognition models has raised significant privacy and bias concerns. Synthetic methods mitigate these concerns and provide scalable and controllable face generation to enable fair and accurate face recognition. However, existing synthetic datasets display limited intraclass and interclass diversity and do not match the face recognition performance obtained using real datasets. Here, we propose VariFace, a two-stage diffusion-based pipeline to create fair and diverse synthetic face datasets to train face recognition models. Specifically, we introduce three methods: Face Recognition Consistency to refine demographic labels, Face Vendi Score Guidance to improve interclass diversity, and Divergence Score Conditioning to balance the identity preservation-intraclass diversity trade-off. When constrained to the same dataset size, VariFace considerably outperforms previous synthetic datasets ($0.9200 \rightarrow 0.9405$) and achieves comparable performance to face recognition models trained with real data (Real Gap = -0.0065). In an unconstrained setting, VariFace not only consistently achieves better performance compared to previous synthetic methods across dataset sizes but also, for the first time, outperforms the real dataset (CASIA-WebFace) across six evaluation datasets. This sets a new state-of-the-art performance with an average face verification accuracy of 0.9567 (Real Gap = +0.0097) across LFW, CFP-FP, CPLFW, AgeDB, and CALFW datasets and 0.9366 (Real Gap = +0.0380) on the RFW dataset.

## 1 Introduction

A decade after the breakthrough performances of DeepFace (Taigman et al., 2014) and DeepID (Sun et al., 2014), deep learning remains the state-of-the-art approach for face recognition (FR) (Wang & Deng, 2021; Gururaj et al., 2024). Deep learning performance is limited by training dataset size (Zhu et al., 2021), and the creation of large-scale FR datasets such as CASIA-WebFace (Yi et al., 2014), MS-Celeb-1M (Guo et al., 2016) and MegaFace (Kemelmacher-Shlizerman et al., 2016) was central to the success of deep learning in FR. However, the development of these massive face datasets involves scraping face image data from the internet without permission from subjects, and all but one of the aforementioned datasets have since been retracted or decommissioned (Boutros et al., 2022). Legally, Article 9 (1) General Data Protection Regulation (GDPR) (Parliament, 2016) in the EU prohibits the processing of biometric data for the purpose of uniquely identifying a natural person, except when the data subject has given explicit consent to the processing of those personal data for one or more specified purposes (Article 9(2)(a) GDPR). Moreover, the AI Act (AIA) in the EU came into effect on August 1st, 2024, where Article 5(1)(e) AIA prohibits the placing on the market or the use of AI systems that create or expand facial recognition databases through the untargeted scraping of facial images (Parliament, 2024). Together, these regulations illustrate a growing legal concern over the use of large-scale web-scraped face datasets for training face recognition models.

Besides privacy concerns, real face datasets suffer from data imbalance, including limited face pose and lighting variation (Liu et al., 2022a), as well as under-representation of protected characteristics such as race and gender (Liu et al., 2022a; Karkkainen & Joo, 2021; Thong et al., 2023). This results in FR models trained on these datasets exhibiting robustness and fairness issues, with the latter raising significant legal and ethical implications (Mehrabi et al., 2021).

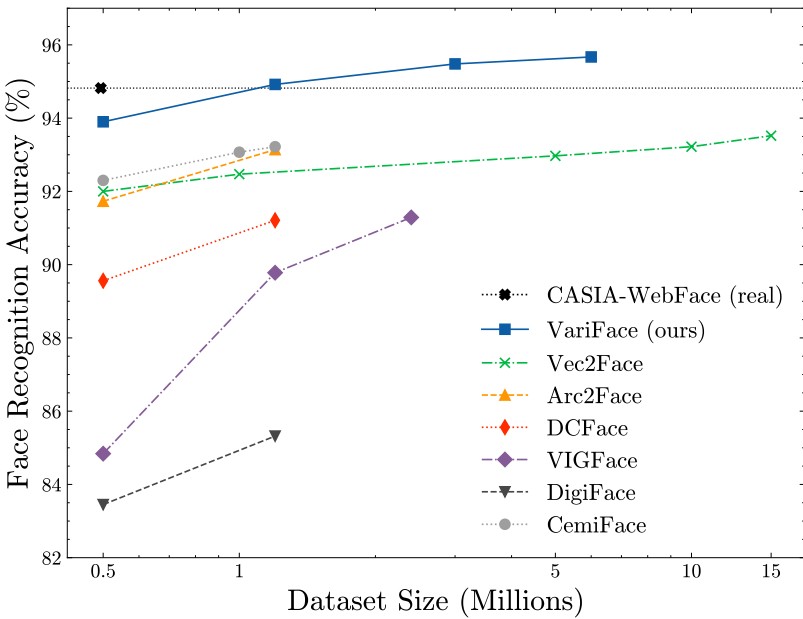

Figure 1: **Face verification accuracy using synthetic datasets.** Face verification accuracy is the average performance across LFW, CFP-FP, CPLFW, AgeDB, and CALFW datasets. VariFace is trained only with data from CASIA-WebFace (real), the performance of which is shown for reference. All other results are taken from their respective papers.

To address the privacy risks and biases associated with real face datasets, there is increasing interest in the development of synthetic face datasets (Baltsou et al., 2024; Qiu et al., 2021; Liu et al., 2022b; Bae et al., 2023; Boutros et al., 2023a; Melzi et al., 2023; Kim et al., 2023). Moreover, synthetic methods not only facilitate scalable dataset generation to support deep FR model training (Trigueros et al., 2021; Bae et al., 2023), but also provide precise control over the composition of the dataset (Liu et al., 2022b; Melzi et al., 2023; Banerjee et al., 2023; Baltsou et al., 2024).

The most widely used synthetic data generation methods are 3D parametric (Wood et al., 2021; Bae et al., 2023) and deep generative (Trigueros et al., 2021; Qiu et al., 2021; Liu et al., 2022b; Boutros et al., 2023b;a; Melzi et al., 2023; Kim et al., 2023) models, the latter encompassing GAN (Deng et al., 2020; Shen et al., 2020; Qiu et al., 2021; Boutros et al., 2022; Liu et al., 2022b; Boutros et al., 2023b; 2024; Wu et al., 2024a) and diffusion-based approaches (Boutros et al., 2023a; Melzi et al., 2023; Kim et al., 2023; Kansy et al., 2023; Papantoniou et al., 2024; Xu et al., 2024). Despite the considerable progress in the development of synthetic face data, there remains a substantial gap in FR performance between models trained on real compared to synthetic data (Boutros et al., 2023c; Shahreza et al., 2024; DeAndres-Tame et al., 2024). This is because synthetic datasets tend to amplify biases inherent in the real face datasets used for training (Thong et al., 2023; Leyva et al., 2024), while also introducing unique challenges that affect FR performance (Bae et al., 2023; Kim et al., 2023).

To mitigate the demographic biases in real datasets, several synthetic methods are able to generate datasets with balanced demographic distributions (Qiu et al., 2021; Kim et al., 2023; Melzi et al., 2023). However, these methods rely on using either existing demographic labels or labels generated by supervised models (Taigman et al., 2014; Kim et al., 2023; Melzi et al., 2023), which exhibit poor performance on minority classes (see Fig. S2). Another issue inherent with synthetic methods is the challenge of balancing identity (ID) preservation with generating diverse images for each individual. Although ID preservation can be achieved using pretrained FR model embeddings as a conditioning signal (Boutros et al., 2023a; Kim et al., 2023; Xu et al., 2024), generating sufficient diversity to reflect 'in-the-wild' variation remains challenging. To generate image variations for an individual, current methods rely on extracting specific attributes such as face pose, expression, and illumination (Melzi et al., 2023; Papantoniou et al., 2024). However, this requires

using supervised models trained on those attributes and ignores properties that cannot be easily classified but are important for FR performance. In summary, while synthetic data generation methods achieve certain aspects, such as balanced demographics and ID preservation, the issue of generating sufficient diversity across and within identities remains unsolved. We hypothesize that this lack of diversity prevents synthetic data from reaching the performance of real data even when scaled significantly (Fig. 1).

To tackle the issue of limited synthetic face diversity, we propose VariFace, a diffusion-based pipeline that achieves fair interclass variation and diverse intraclass variation. To obtain accurate demographic labels, we first leverage a pretrained CLIP encoder (Radford et al., 2021) to extract initial predictions. These labels are subsequently refined with dataset-level information by enforcing consistency within FR embedding space. The race and gender labels are used as conditioning signals for the first stage to generate a demographically-balanced dataset of face identities. Moreover, interclass diversity is further improved by applying the Vendi score (Friedman & Dieng, 2023) as a guidance loss function during sampling. To generate diverse intraclass variation, we propose Divergence Score Conditioning, a metric in the FR embedding space that enables control over the ID preservation-intraclass diversity trade-off. This avoids the need to manually specify image attributes, greatly simplifying the generation pipeline by avoiding the use of auxiliary supervised models. By conditioning with divergence score, ID, and age labels, the second stage generates diverse but ID-preserved variation across individuals.

While the use of web-scraped face datasets should be avoided when training FR models, for the purpose of benchmarking, we follow previous methods and train VariFace on the CASIA-WebFace dataset. When constrained to the same dataset size, our proposed approach achieves comparable performance to state-of-the-art FR models trained with real face data. Moreover, by scaling dataset size, we demonstrate for the first time face verification accuracy that exceeds the real dataset used for training and achieve a new state-of-the-art across six evaluation datasets (Fig. 1).

In this paper, we propose the following contributions:

- We propose a two-stage, diffusion-based face generation pipeline that achieves fair interclass variation and diverse intraclass variation.

- We introduce Face Recognition Consistency to refine demographic labels, Face Vendi Score Guidance to improve interclass diversity, and Divergence Score Conditioning to control the ID preservation-intraclass diversity trade-off.

- We achieve a new state-of-the-art FR performance across six face evaluation datasets using only synthetic data, outperforming previous synthetic methods and the real dataset used for training.

## 2 Related work

**Synthetic Face Generation**. The applications of generating synthetic faces are extensive and include face recognition (Boutros et al., 2023c; Bae et al., 2023; Boutros et al., 2023a; Melzi et al., 2023; Kim et al., 2023; Kansy et al., 2023), reconstruction (Richardson et al., 2016; 2017), editing (Shen et al., 2020; Matsunaga et al., 2022; Plesh et al., 2023; Wu et al., 2024b) and analysis (Li & Deng, 2020; Wood et al., 2021).

Using 3D face scan data, Wood et al. (2021) applied a face model with a library of hair, clothing, and accessory assets to render a face dataset. 3D scan data avoids the need to use real face data while maintaining the generation of well-preserved identities. However, the acquisition of 3D face scans is time-consuming and expensive due to the need for specialized scanner equipment and considerable post-processing requirements. Moreover, there remains a significant domain gap between rendered and real faces, and the variation is limited by the assets available.

In contrast, deep generative methods such as GANs and diffusion models address domain gap issues by leveraging real datasets to learn the generation of realistic faces. DiscoFaceGAN (Deng et al., 2020) combines 3D priors with an adversarial learning framework to generate faces with independent control over ID, pose, expression, and illumination. Similarly, InterFaceGAN (Shen et al., 2020) enables precise control of facial attributes while preserving ID by manipulating the latent representation along semantically meaningful

directions. SynthDistill (Shahreza et al., 2023) combines StyleGAN generation with a knowledge distillation framework to dynamically sample hard examples during training. Despite high controllability, GANs suffer from mode collapse, resulting in limited interclass and intraclass diversity. In contrast, diffusion models facilitate diverse face image generation while retaining capabilities for fine-grained, controllable face editing (Matsunaga et al., 2022; Kim et al., 2023).

**Synthetic Datasets for Face Recognition**. There is a growing interest in applying synthetic datasets for training FR models, with a surge in related competitions held in recent years (Shahreza et al., 2024; DeAndres-Tame et al., 2024; Melzi et al., 2024). Synthetic data provides a cost-effective method to scale datasets, facilitating the training of deep FR models that require many unique faces with a diverse set of images per individual (Zhu et al., 2021; An et al., 2022).

Based on 3D face scan data, the DigiFace-1M (Bae et al., 2023) dataset comprises over a million rendered faces. While 3DCG data avoids privacy concerns and issues with ID preservation, the domain gap and low intraclass diversity limit FR performance (Rahimi et al., 2024). In contrast, deep generative models suffer from problems with ID preservation and image artifacts but offer more flexibility to generate diversity across and within individuals. SynFace (Qiu et al., 2021) leverages DiscoFaceGAN for face generation and introduces ID mixup to improve interclass diversity. However, SynFace suffers from low intraclass diversity, and IDiff-Face (Boutros et al., 2023a) applies dropout during training to prevent overfitting to ID context and improve intraclass diversity. Similarly, to enhance intraclass diversity, DCFace (Kim et al., 2023) uses a bank of real images for style information. However, there is a risk of data leakage using real images for style transfer, and these methods may not help to improve diversity beyond the training dataset. Instead, GANDiffFace (Melzi et al., 2023) uses supervised labels from pretrained models to generate individuals of diverse ages, as well as the same individual with diverse poses and expressions. Similarly, ID$^3$ (Xu et al., 2024) uses age and pose labels as a face attribute conditioning signal to train a conditional diffusion model. While supervised attributes can further improve intraclass diversity, the limited attributes specified may not capture all the variations important for FR. The most recent methods, such as VIGFace (Kim et al., 2024), Arc2Face (Papantoniou et al., 2024), Vec2Face (Wu et al., 2024a) and CemiFace (Sun et al., 2024) focus on generating ID embeddings to synthesize diverse face images. However, these methods do not address fairness concerns, inheriting demographic biases from the real dataset (see Sec. 4.5), and there remains a considerable gap between the performance of FR models trained on these datasets compared to real datasets.

## 3 Method

VariFace is a two-stage, diffusion-based pipeline for synthetic face dataset generation. The training and inference pipeline is summarized in Fig. 2.

### 3.1 Stage 1: Fair interclass variation

The aim of the first stage of the pipeline is to generate a diverse set of face identities with a balanced representation of races and genders. To create demographic labels, we first obtain initial race and gender labels using a pretrained CLIP model (Radford et al., 2021; Wang et al., 2023), and subsequently refine these predictions using a pretrained FR model. With these labels, we train a conditional diffusion model (Ho et al., 2020; Nichol & Dhariwal, 2021; Dhariwal & Nichol, 2021; Rombach et al., 2022; Crowson et al., 2024) to learn to generate faces with control over race and gender attributes. During sampling, we apply the Vendi score (Friedman & Dieng, 2023) as a guidance loss function to generate diversity within each demographic category. Finally, we apply an automatic filtering process to retain images that are demographic-consistent and display good face image quality for use as synthetic identities in the second stage.

**Label refinement with Face Recognition Consistency**. Previous methods used supervised approaches such as DeepFace (Taigman et al., 2014) to extract demographic labels from face images (Melzi et al., 2023; Karkkainen & Joo, 2021). Instead, we leverage a pretrained CLIP model to obtain race, gender, and age labels (see Suppl. Sec. A). While DeepFace uses separate age, race, and gender models, CLIP benefits from a unified embedding space and offers greater flexibility in defining labels. However, both supervised methods and CLIP generate predictions independently for each face image, ignoring crucial contextual information

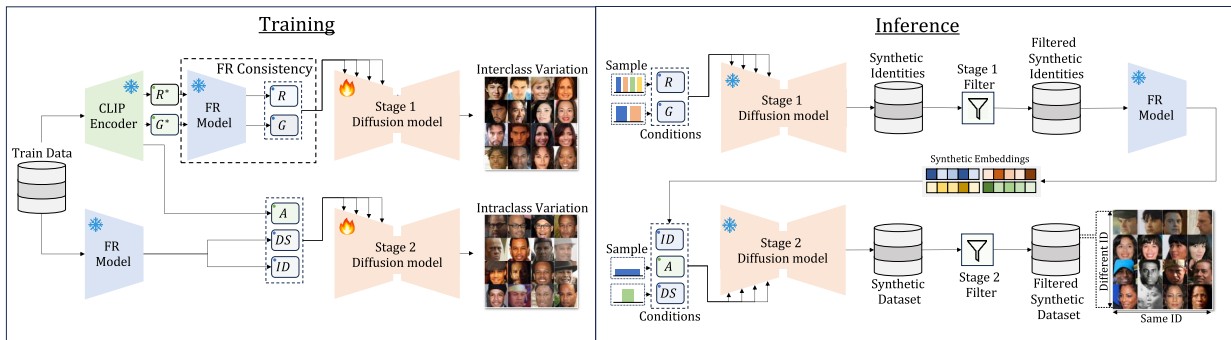

Figure 2: **VariFace Training and Inference Pipeline.** Training: Predictions for race (R*), gender (G*), and age (A) are extracted using a pretrained CLIP model. Next, a pretrained FR model is used to refine race (R) and gender (G) labels, as well as compute identity (ID) embeddings and divergence scores (DS). These labels are used to train conditional diffusion models to generate interclass and intraclass variation in stage 1 and 2, respectively. Inference: The stage 1 diffusion model generates a balanced dataset of synthetic identities, which are subsequently filtered and processed with a pretrained FR model to generate a set of synthetic embeddings. The synthetic ID embeddings and randomly sampled A and DS are used as conditions for the stage 2 diffusion model to generate a synthetic face dataset, which is passed through the second stage filter to create the filtered synthetic dataset.

present in the dataset. Face embeddings extracted using FR models not only contain ID information but also the space of face embeddings appears well structured with respect to demographic information (Li et al., 2023; Leyva et al., 2024). Therefore, we propose to leverage the structure in the FR embedding space to refine the race and gender labels.

Specifically, let $I$ represent an image from the training face dataset $D$. We first transform $I$ using a pretrained FR model to obtain a face embedding, $E \in \mathbb{R}^n$. Given two face embeddings $E_i$ and $E_j$, where $i, j \in D$, we compute the cosine similarity ($C_S$):

$$C_S(E_i, E_j) = \frac{E_i \cdot E_j}{\|E_i\|\|E_j\|}. \tag{1}$$

For each embedding, we select the top $K$ similar embeddings from $D$ measured by $C_S$. Using this embedding subset, we redefine the demographic label belonging to the query image as the most frequent label associated with the top $K$ similar subset of face embeddings. Accordingly, combining CLIP and Face Recognition Consistency (CLIP-FRC) provides synergistic representations to obtain accurate demographic labels (see Fig. S2).

**Enhancing diversity with Face Vendi Score Guidance**. Generating a diverse set of face identities is important for training FR models (Kim et al., 2023). The Vendi score (VS) is an evaluation metric that measures diversity at a dataset level and is defined as the exponential of the Shannon entropy applied to the normalized eigenvalues of a kernel similarity matrix.

Formally, given a dataset $X = \{x_i\}_{i=1}^n$ with domain $\mathcal{X}$, a positive semidefinite similarity function $k : \mathcal{X} \times \mathcal{X} \to \mathbb{R}$ and its associated Kernel matrix $K \in \mathbb{R}^{n \times n}$ with $K_{i,j} = k(x_i, x_j)$, the VS is defined as:

$$VS(X; k) = \exp \left( -\sum_{i=1}^n \overline{\lambda}_i \log \overline{\lambda}_i \right), \tag{2}$$

where $\{\overline{\lambda}_i\}_{i=1}^n$ are the normalized eigenvalues of $K$ such that $\overline{\lambda}_i = \frac{\lambda_i}{\sum_{i=1}^n \lambda_i}$.

Here, we propose using the VS as a guidance function to improve interclass diversity. In contrast to conditional image generation (Ho & Salimans, 2022; Bansal et al., 2023), guided image generation involves using a pretrained frozen diffusion model to control image generation (Kim et al., 2022a; Bansal et al., 2024). By specifying a guidance loss function, the sampling process can be guided to simultaneously optimize the guidance loss function and denoising objective. Concretely, for each batch of denoised images, we obtain face embeddings $E \in \mathbb{R}^n$ using a pretrained FR model and compute the VS loss ($\mathcal{L}_{VS}$) specifying the dataset as the batch of embeddings $B = \{E\}_{i=1}^m$ and similarity function as the cosine similarity $C_S : \mathbb{R}^n \times \mathbb{R}^n \rightarrow \mathbb{R}$ defined in Eq. 1:

$$\mathcal{L}_{VS} = -VS(B; C_S). \tag{3}$$

The Face Vendi Score Guidance algorithm for a generic diffusion sampler $S(\cdot, \cdot, \cdot)$ is summarized in Algorithm 1. Similar to Universal Guidance (Bansal et al., 2024), we apply the guidance loss to the denoised image. Specifically, we use a pretrained FR model on the batch of denoised images to generate a set of face embeddings as input into $\mathcal{L}_{VS}$. The gradient of $\mathcal{L}_{VS}$ is then scaled and combined with the noise estimated by the diffusion model and used by the diffusion sampler to generate the prediction for the following timestep. By maximizing the Vendi Score loss across timesteps, the batch of faces generated is guided to become diverse with respect to face ID, therefore maximizing interclass diversity.

---

**Algorithm 1** Face Vendi Score Guidance

---

**Input:** Batch of image latent vectors $z_t$, diffusion model $\epsilon_\theta$, FR model $f_\phi$, noise scale $\alpha_t$, Vendi Score loss function $\mathcal{L}_{VS}$, guidance scale $s$, timesteps $t = 0, 1, ..., T$, race condition $r$, gender condition $g$, general sampling function $S(\cdot, \cdot, \cdot)$

**Output:** $z_{t-1}$

    **for** $t = T$ to $t = 1$ **do**,

        $\hat{z}_0 \leftarrow \frac{z_t - (\sqrt{1-\alpha_t})\epsilon_\theta(z_t, t, r, g)}{\sqrt{\alpha_t}}$

        $\hat{E}_0 \leftarrow f_\phi(\hat{z}_0)$

        $\hat{\epsilon}_\theta(z_t, t, r, g) \leftarrow \epsilon_\theta(z_t, t, r, g) + s \cdot \nabla_{z_t} \mathcal{L}_{VS}(\hat{E}_0)$

        $z_{t-1} \leftarrow S(z_t, \hat{\epsilon}_\theta, t)$

        $\epsilon' \sim \mathcal{N}(0, I)$

        $z_t \leftarrow \sqrt{\alpha_t/\alpha_{t-1}} z_{t-1} + \sqrt{(1 - \alpha_t/\alpha_{t-1})}\epsilon'$

    **end for**

---

**Filtering**. To create a balanced, high-quality set of face identities, we apply a two-stage filtering process to the generated images. Firstly, we obtain demographic labels using CLIP-FRC and remove images that are inconsistent with the intended label. Secondly, to remove images with extreme poses, poor lighting, or artifacts, we use CLIB-FIQA (Ou et al., 2024) to extract face image quality labels and remove images with a score below a predefined threshold.

## 3.2 Stage 2: Diverse intraclass variation

The second stage creates diverse image variations for each face generated in the first stage while preserving face ID. We train the second diffusion model conditioned on face identities, ages, and divergence scores obtained from the training dataset. To obtain face identities, we first extract face embeddings using a pretrained FR model and then compute the mean embedding for each ID label, normalizing the embeddings to a unit sphere. The age and divergence scores are obtained using a pretrained CLIP and FR model, respectively. At inference, we use the face identities obtained from stage 1, and for each ID, we randomly sample the age and the divergence scores to generate diverse intraclass variation. To reduce label noise, the final filtering process ensures the generated images preserve face ID.

**Divergence score conditioning**. To generate diverse variation for a given individual, it is common to specify a set of attributes such as pose or lighting, either using text prompts (Baltsou et al., 2024), or labels derived from classifiers (Melzi et al., 2023). However, these methods are non-exhaustive and may struggle

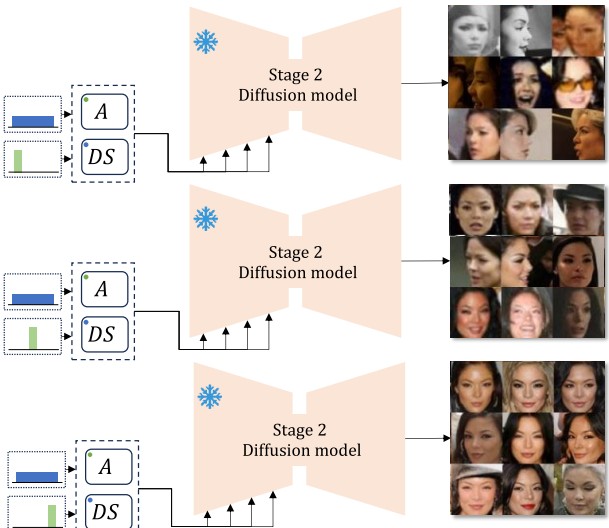

Figure 3: **Divergence Score Conditioning.** By varying the divergence scores applied during sampling, the diversity in generated images can be controlled. From top to bottom, the DS values used are 0.4, 0.6, and 0.8, respectively. All the images are derived from the same synthetic identity.

to capture variations that cannot be easily conveyed in natural language or classified. Instead, we aim to capture diversity more holistically and without the reliance on named attributes, greatly simplifying the generation pipeline by removing the need for additional models. To do so, we assign a score to each training image based on how far the image deviates from the 'prototypical example' for that individual and use this score as a label for training a conditional diffusion model.

Specifically, suppose $X_{i,j}$ denotes the $j$-th sample of the $i$-th ID in the training dataset $X$. We first obtain the corresponding face embedding $E_{i,j} \in \mathbb{R}^n$ with a pretrained FR model. The 'prototypical example' for the $i$-th ID is defined as its mean embedding over all samples for that ID:

$$\overline{E}_i = \frac{1}{|E_i|} \sum_j E_{i,j}. \tag{4}$$

Then, for the image with the associated embedding $E_{i,j}$, we define its divergence score (DS) as:

$$DS(E_{i,j}) = C_S(E_{i,j}, \overline{E}_i). \tag{5}$$

At inference, the divergence score condition can be used to control the intraclass diversity (Fig. 3).

**Filtering**. There is an inherent trade-off between intraclass diversity and ID preservation (Kim et al., 2023). While controlling the DS helps to balance the trade-off, we further enforce this by applying a filter to remove cases where the ID was not preserved. Using the pretrained FR model, we compute the cosine similarity between the embedding from the base image generated in stage 1, with the images generated in stage 2 using that ID as a condition. We then remove images where the cosine similarity between the embeddings is below a predefined threshold.

# 4 Experiments

## 4.1 Implementation details

For the diffusion model, we adapted the Hourglass Diffusion Transformer (HDiT) (Crowson et al., 2024) architecture to handle multiple conditions (see Fig. S7). We trained the stage 1 and 2 diffusion models for 700K and 3M iterations, which took ≈19 hours and 3.5 days, respectively, using 4 NVIDIA A100 GPUs.

| Dataset | Size (ID × images/ID) | LFW | CFP-FP | CPLFW | AgeDB | CALFW | Average | Real Gap |
|---|---|---|---|---|---|---|---|---|
| CASIA-WebFace | 0.5M ($\approx 10.5K \times 47$) | 0.9950 | 0.9536 | 0.9007 | 0.9487 | 0.9368 | 0.9470 | 0.0000 |
| SynFace (Qiu et al., 2021) | 0.5M ($10K \times 50$) | 0.8407 | 0.6337 | 0.6355 | 0.5910 | 0.6937 | 0.6789 | -0.2681 |
| IDNet* (Kolf et al., 2023) | 0.5M ($10K \times 50$) | 0.9258 | 0.7540 | 0.7425 | 0.6388 | 0.7990 | 0.7913 | -0.1557 |
| ExFaceGAN* (Boutros et al., 2023b) | 0.5M ($10K \times 50$) | 0.9350 | 0.7384 | 0.7160 | 0.7892 | 0.8298 | 0.8017 | -0.1453 |
| DigiFace* (Bae et al., 2023) | 0.5M ($10K \times 50$) | 0.9540 | 0.8740 | 0.7887 | 0.7697 | 0.7862 | 0.8345 | -0.1125 |
| VIGFace* (Kim et al., 2024) | 0.5M ($10K \times 50$) | 0.9660 | 0.8666 | 0.7503 | 0.8250 | 0.8342 | 0.8484 | -0.0986 |
| IDiff-Face* (Boutros et al., 2023a) | 0.5M ($10K \times 50$) | 0.9800 | 0.8547 | 0.8045 | 0.8643 | 0.9065 | 0.8820 | -0.0650 |
| DCFace* (Kim et al., 2023) | 0.5M ($10K \times 50$) | 0.9855 | 0.8533 | 0.8262 | 0.8970 | 0.9160 | 0.8956 | -0.0514 |
| ID³* (Xu et al., 2024) | 0.5M ($10K \times 50$) | 0.9768 | 0.8684 | 0.8277 | 0.9100 | 0.9073 | 0.8980 | -0.0490 |
| Arc2Face* (Papantoniou et al., 2024) | 0.5M ($10K \times 50$) | 0.9881 | 0.9187 | 0.8516 | 0.9018 | 0.9263 | 0.9173 | -0.0297 |
| Vec2Face* (Wu et al., 2024a) | 0.5M ($10K \times 50$) | 0.9887 | 0.8897 | 0.8547 | 0.9312 | **0.9357** | 0.9200 | -0.0270 |
| CemiFace* Sun et al. (2024) | 0.5M ($10K \times 50$) | 0.9903 | 0.9106 | 0.8765 | 0.9133 | 0.9242 | 0.9230 | -0.0228 |
| VariFace (ours) | 0.5M ($10K \times 50$) | **0.9938** | **0.9460** | **0.8882** | **0.9438** | 0.9305 | **0.9405** | **-0.0065** |

Table 1: **Face Verification Accuracy with constrained synthetic dataset size evaluated on the Standard Benchmark.** *Results taken from original papers. The best performance is highlighted in bold for each evaluation dataset, and the second-best performance is underlined.

During inference, we applied the DPM-Solver++(3M) SDE sampler (Lu et al., 2022; Crowson et al., 2024) with 50 sampling steps. To create demographic labels, we used the pretrained ViT-L-14 MetaCLIP model (Xu et al., 2023). For FR consistency, we used an IResNet-100 model trained on the CASIA-WebFace dataset to extract embeddings and set $K = 50$. Regarding filtering settings, we empirically set the quality threshold in stage 1 at 0.7 and the cosine similarity threshold at 0.3 in stage 2 (Wu et al., 2024a). For the stage 1 generation, we sampled a balanced set of individuals across races and genders. For the stage 2 generation, we uniformly sampled age values $A \sim \mathcal{U}(0, 1)$ and divergence scores $DS \sim \mathcal{U}(0.5, 0.8)$ (see Table S2).

Within the synthetic generation pipeline, where performance is the main concern, we used an IResNet-100 model, while for evaluation, we used the smaller IResNet-50 architecture. For both models, we used the ArcFace loss function (Deng et al., 2019) and trained for 40 epochs, with an initial learning rate of 0.1, which is reduced by a factor of 10 at epochs 24, 30, and 36. Moreover, we applied the following data augmentations: horizontal flip, sharpness, contrast, equalization, and random erasing. More detailed information on the diffusion and FR model settings can be found in the Suppl. Sec. F and G, respectively.

## 4.2 Datasets

For training, we used the CASIA-WebFace dataset (Yi et al., 2014), a publicly available face dataset containing 490,414 images from 10,575 individuals. For evaluation, we selected six common face verification datasets: LFW (Huang et al., 2008), CFP-FP (Sengupta et al., 2016), CPLFW (Zheng & Deng, 2018), AgeDB (Moschoglou et al., 2017), CALFW (Zheng et al., 2017) and RFW dataset (Wang et al., 2019). These datasets were designed to evaluate specific aspects of FR performance, including pose variation (CFP-FP, CPLFW), large age differences (AgeDB, CALFW), and race (RFW). Following previous methods (Kim et al., 2023; 2024; Wu et al., 2024a), we group the LFW, CFP-FP, CPLFW, AgeDB, and CALFW datasets and refer to these as the 'Standard Benchmark', and we report the performance difference to a baseline model trained on CASIA-WebFace (Real Gap).

## 4.3 Constrained Face Recognition Results

The face verification performance on the Standard Benchmark, using different synthetic datasets constrained to the same dataset size and images per ID, are shown in Table 1. When constrained to the same dataset size as the real dataset, no synthetic method outperformed the real dataset. Compared to previous synthetic methods, our proposed method consistently achieves the best performance, with a considerable improvement over previous state-of-the-art $(0.9230 \rightarrow 0.9405)$ and the smallest overall performance gap compared to the real dataset (Real Gap $= -0.0065$).

The performance on the RFW dataset using a constrained dataset size and images per ID is shown in Table 2. Similarly, when constrained to the same dataset size and images per ID, the performance using synthetic datasets cannot match the overall performance obtained using real data. However, VariFace con-

| Dataset | African | Asian | Caucasian | Indian | Average | Real Gap |
|---|---|---|---|---|---|---|
| CASIA-WebFace | 0.8822 | 0.8697 | 0.9448 | 0.8978 | 0.8986 | 0.0000 |
| SynFace (Qiu et al., 2021) | 0.5643 | 0.6355 | 0.6647 | 0.6457 | 0.6276 | -0.2710 |
| DigiFace (Bae et al., 2023) | 0.5952 | 0.6408 | 0.6750 | 0.6427 | 0.6384 | -0.2602 |
| DCFace (Kim et al., 2023) | 0.7742 | 0.8122 | 0.8917 | 0.8460 | 0.8310 | -0.0676 |
| Vec2Face (Wu et al., 2024a) | 0.8415 | 0.8535 | 0.9028 | 0.8750 | 0.8682 | -0.0304 |
| VariFace (ours) | **0.8895** | **0.8733** | **0.9295** | **0.8988** | **0.8978** | **-0.0008** |

Table 2: **Face Verification Accuracy with constrained synthetic dataset size evaluated on the RFW dataset.** All datasets contain 0.5M images, with 50 images per ID. The best performance is highlighted in bold for each dataset, and the second-best performance is underlined.

| Dataset | Size (ID × images/ID) | LFW | CFP-FP | CPLFW | AgeDB | CALFW | Average | Real Gap |
|---|---|---|---|---|---|---|---|---|
| CASIA-WebFace | 0.5M ($\approx 10.5K \times 47$) | 0.9950 | 0.9536 | 0.9007 | 0.9487 | 0.9368 | 0.9470 | 0.0000 |
| SynFace (Qiu et al., 2021) | 1.0M ($10K \times 100$) | 0.8580 | 0.6473 | 0.6395 | 0.5765 | 0.6987 | 0.6840 | -0.2630 |
| DigiFace* (Bae et al., 2023) | 1.2M ($10K \times 72 + 100K \times 5$) | 0.9582 | 0.8877 | 0.8162 | 0.7972 | 0.8070 | 0.8532 | -0.0938 |
| DCFace* (Kim et al., 2023) | 1.2M ($20K \times 50 + 40K \times 5$) | 0.9858 | 0.8861 | 0.8507 | 0.9097 | 0.9282 | 0.9121 | -0.0349 |
| VIGFace* (Kim et al., 2024) | 4.2M ($60 \times 50 + 60 \times 20$) | 0.9913 | 0.9187 | 0.8483 | 0.9463 | 0.9338 | 0.9277 | -0.0193 |
| Arc2Face* (Papantoniou et al., 2024) | 1.2M ($20K \times 50 + 40K \times 5$) | 0.9892 | 0.9458 | 0.8645 | 0.9245 | 0.9333 | 0.9314 | -0.0156 |
| CemiFace* (Sun et al., 2024) | 1.2M ($20K \times 50 + 40K \times 5$) | 0.9922 | 0.9284 | 0.8886 | 0.9213 | 0.9303 | 0.9322 | -0.0148 |
| Vec2Face* (Wu et al., 2024a) | 15M ($300K \times 50$) | 0.9930 | 0.9154 | 0.8770 | 0.9445 | 0.9458 | 0.9352 | -0.0118 |
| VariFace (ours) | 0.5M ($25K \times 20$) | 0.9938 | 0.9407 | 0.8930 | 0.9417 | 0.9357 | 0.9410 | -0.0060 |
| VariFace (ours) | 1.2M ($60K \times 20$) | 0.9945 | 0.9561 | 0.9063 | 0.9475 | 0.9413 | 0.9492 | +0.0022 |
| VariFace (ours) | 3.0M ($60K \times 50$) | 0.9950 | 0.9609 | 0.9145 | **0.9593** | 0.9445 | 0.9548 | +0.0078 |
| VariFace (ours) | 6.0M ($60K \times 100$) | **0.9960** | **0.9637** | **0.9205** | 0.9568 | **0.9467** | **0.9567** | +0.0097 |

Table 3: **Face Verification Accuracy with unconstrained synthetic dataset size evaluated on the Standard Benchmark.** *The best results are taken from the original papers. For each dataset, the best performance is highlighted in bold, and the second-best performance is underlined.

siderably outperforms previous methods (0.8682 → 0.8978) and achieves comparable performance to real data (Real Gap = −0.0008). Importantly, VariFace outperforms the real dataset across all minority race categories: African (0.8822 → 0.8895), Asian (0.8697 → 0.8733) and Indian (0.8978 → 0.8988).

### 4.4 Unconstrained Face Recognition Results

One of the main benefits of using synthetic data is the ease of scaling the dataset size, involving either synthesizing new IDs or increasing the number of images per ID. The face verification performance on Standard Benchmark, using different synthetic datasets without dataset size constraint, is shown in Table 3.

The best performance was obtained using VariFace, achieving an average accuracy of 0.9567 at 6M images, outperforming the real dataset (Real Gap = +0.0097). In fact, VariFace outperforms real data at 1.2M images (0.9470 → 0.9492). In contrast, no other synthetic method at any dataset size reaches real data performance. We observe that either increasing the number of IDs ($25K \rightarrow 60K$) or increasing the number of images per ID ($20 \rightarrow 100$) improves performance across evaluation datasets.

| Dataset | Size (ID × images/ID) | African | Asian | Caucasian | Indian | Average | Real Gap |
|---|---|---|---|---|---|---|---|
| CASIA-WebFace | 0.5M ($\approx 10.5K \times 47$) | 0.8822 | 0.8697 | 0.9448 | 0.8978 | 0.8986 | 0.0000 |
| SynFace (Qiu et al., 2021) | 1.0M ($10K \times 100$) | 0.5873 | 0.6617 | 0.6758 | 0.6502 | 0.6438 | -0.2548 |
| DigiFace (Bae et al., 2023) | 1.2M ($10K \times 72 + 100K \times 5$) | 0.6145 | 0.6648 | 0.6910 | 0.6603 | 0.6577 | -0.2409 |
| DCFace (Kim et al., 2023) | 1.2M ($20K \times 50 + 40K \times 5$) | 0.8157 | 0.8345 | 0.9083 | 0.8765 | 0.8588 | -0.0398 |
| Vec2Face (Wu et al., 2024a) | 1.0M ($20K \times 50$) | 0.8763 | 0.8695 | 0.9185 | 0.8962 | 0.8901 | -0.0085 |
| VariFace (ours) | 0.5M ($25K \times 20$) | 0.8973 | 0.8825 | 0.9277 | 0.9042 | 0.9029 | +0.0043 |
| VariFace (ours) | 1.2M ($60K \times 20$) | 0.9130 | 0.9008 | 0.9445 | 0.9172 | 0.9189 | +0.0203 |
| VariFace (ours) | 3.0M ($60K \times 50$) | 0.9292 | 0.9130 | 0.9560 | 0.9257 | 0.9310 | +0.0324 |
| VariFace (ours) | 6.0M ($60K \times 100$) | **0.9363** | **0.9180** | **0.9590** | **0.9332** | **0.9366** | +0.0380 |

Table 4: **Face Verification Accuracy with unconstrained synthetic dataset size evaluated on the RFW dataset.** For each dataset, the best performance is highlighted in bold, and the second-best performance is underlined.

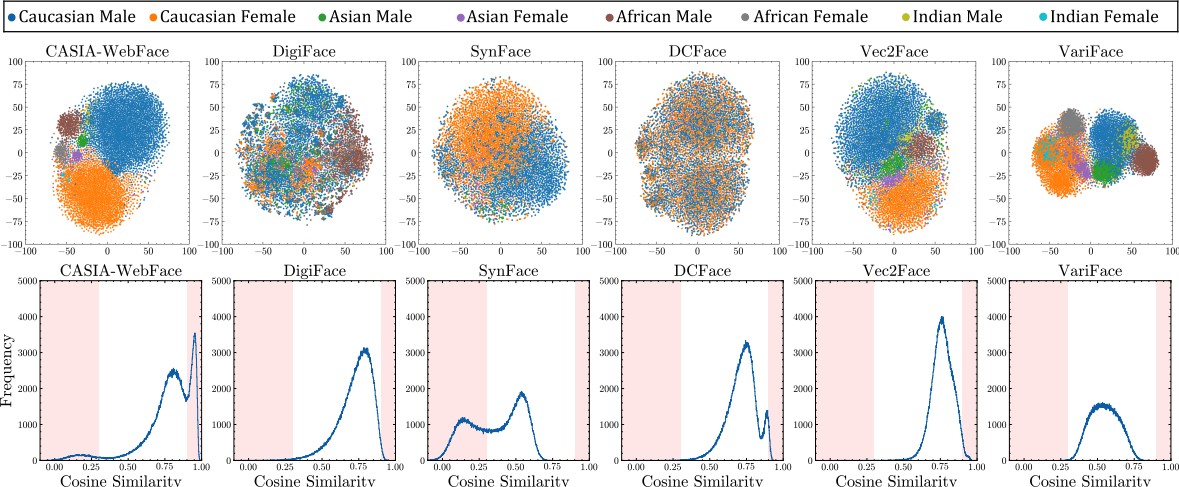

Figure 4: **Synthetic dataset characteristics.** Top: t-SNE plots of mean face embeddings for identities in different synthetic datasets. The race and gender labels for each embedding are represented by different colors defined in the legend. Bottom: Histogram of divergence scores for different synthetic datasets. The regions where cosine similarity score < 0.3 and > 0.9 are shaded in red. CASIA-WebFace (real dataset) is shown for reference.

The performance on the RFW dataset using an unconstrained dataset is shown in Table 4. Similarly, the best performance on the RFW dataset was observed with VariFace, achieving an average score of 0.9366 at 6M images, outperforming real data (Real Gap = +0.0380). Importantly, VariFace outperforms real data at the same dataset size of 0.5M when the images per ID are unconstrained (Real Gap = +0.0043). In contrast, no other synthetic method achieves the same performance as real data at the dataset sizes evaluated.

### 4.5 Synthetic dataset characteristics

We analyze the characteristics of the VariFace and open-source synthetic 0.5M datasets with respect to fairness and intraclass diversity.

**Fairness**. t-SNE plots of different synthetic datasets are shown in Fig. 4. The t-SNE plot of CASIA-WebFace identities shows large clusters corresponding to Caucasian males and females. In contrast, while there are smaller clusters for African and Asian races, there are no clusters for Indian individuals. Vec2Face demonstrates clusters similar to those in CASIA-WebFace, inheriting the demographic biases from the real dataset. DigiFace displays multiple but indistinct clusters for each race, while only the Caucasian clusters are retained using SynFace, and no clusters are observed with DCFace. In contrast, VariFace reveals clusters for each race and gender category, including for Indian individuals, which were not observed in the real dataset.

**Intraclass diversity**. Histogram plots of DS for different synthetic datasets are shown in Fig. 4. Low cosine similarity values generally correspond to label noise, representing either mislabeled data for real datasets or lack of ID preservation for synthetic datasets (see Suppl. Sec. C). Among the synthetic datasets, SynFace displays many images with low cosine similarity values due to the use of ID mixing (Qiu et al., 2021). In contrast, other synthetic methods contain mainly high cosine similarity values, corresponding to low intraclass diversity. VariFace controls the DS to enable the generation of diverse image variation while preserving ID.

### 4.6 Dataset generation time

In addition to performance improvements, VariFace also takes less time to generate face datasets than previous methods (Table 5).

| Method | (↓) Dataset Generation Time (hours) |
|---|---|
| DCFace (Kim et al., 2023) | 20 |
| Vec2Face (Wu et al., 2024a) | 36 |
| VariFace (ours) | **12** |

Table 5: **Dataset generation time.** 500K images generated using second stage generative model on a single NVIDIA A100 GPU.

| Demographic | Age | Divergence | LFW | CFP-FP | CPLFW | AgeDB | CALFW | Average | African | Asian | Caucasian | Indian | Average |
|---|---|---|---|---|---|---|---|---|---|---|---|---|---|
| ✓ | ✗ | ✗ | 0.9903 | 0.9267 | 0.8672 | 0.9200 | 0.9148 | 0.9238 | 0.8662 | 0.8580 | 0.9117 | 0.8683 | 0.8760 |
| ✓ | ✓ | ✗ | 0.9927 | 0.9274 | 0.8735 | 0.9295 | 0.9338 | 0.9314 | 0.8757 | 0.8652 | 0.9237 | 0.8870 | 0.8879 |
| ✓ | ✗ | ✓ | 0.9917 | 0.9447 | 0.8882 | 0.9273 | 0.9173 | 0.9338 | 0.8812 | **0.8737** | 0.9252 | 0.8793 | 0.8898 |
| ✗ | ✓ | ✓ | **0.9948** | 0.9444 | **0.8962** | 0.9403 | 0.9295 | **0.9411** | 0.8650 | 0.8365 | **0.9355** | 0.8912 | 0.8820 |
| ✓ | ✓ | ✓ | 0.9938 | **0.9460** | 0.8882 | **0.9438** | **0.9305** | 0.9405 | **0.8895** | 0.8733 | 0.9295 | **0.8988** | **0.8978** |

Table 6: **Conditioning ablation study.** Face verification accuracy using VariFace with different combinations of conditioning signals. Demographic = race and gender conditioning in stage 1. Age = age condition in stage 2. Divergence = divergence score condition in stage 2. All datasets contain 0.5M images (50 images/ID).

The entire pipeline to generate 500K images using VariFace, including ID generation and filtering, took ≈15 hours. There are three main reasons for Variface's efficiency. Firstly, while FVSG adds inference overhead to stage 1 generation, this cost is minimal because face datasets contain significantly fewer IDs than images. Therefore, the stage 1 generation is fast, taking ≈15 minutes to generate 10,000 IDs without FVSG, or ≈40 minutes with FVSG. The main bottleneck is in the second stage, where multiple images of the same ID are generated. To minimize inference costs in this stage, our proposed DSC provides a lightweight solution to generate diversity without relying on additional gradient computations through auxiliary models. Finally, we adopt an efficient diffusion architecture and fast sampling method that reduces inference time across both stages.

### 4.7 Ablation study

**Conditioning signal**. To assess the effect of each conditioning signal used in VariFace, we evaluate the FR performance trained with VariFace using different combinations of conditions (Table 6). By removing the race and gender conditioning in stage 1, the performance on the Standard Benchmark remains comparable (0.9405 → 0.9411), but there is a considerable decrease in RFW performance (0.8978 → 0.8820), especially with minority races (Asian: 0.8733 → 0.8365, Indian: 0.8988 → 0.8683, African: 0.8895 → 0.8662). Both age and DS conditioning improve face verification accuracy across all datasets, with age conditioning improving performance on the AgeDB and CALFW datasets, and DS conditioning improving performance on the CFP-FP and CPLFW datasets. Overall, the best performance is achieved using all three conditions, providing fairness across races and robustness to significant pose and age variation.

**Face Vendi Score Guidance**. The purpose of FVSG is to improve the interclass diversity (see Suppl Sec. B). Previous methods to improve interclass diversity relied on ID filtering (Kim et al., 2023; Wu et al., 2024a), where FR embeddings from synthesized identities with a cosine similarity above a predefined threshold are removed. Here, we demonstrate that FVSG improves face verification accuracy and is a better alternative to ID filtering (Table 7), with the most improvement observed on the RFW dataset (0.8935 → 0.8978). In contrast, we observe that ID filtering was associated with a decrease in performance across datasets (0.8935 → 0.8927). While ID filtering and FVSG are designed for the same purpose, ID filtering does not encourage sampling more diverse IDs and is sensitive to the FR model and threshold used. Moreover, the removal of similar IDs using ID filtering risks removing difficult-to-classify but distinct IDs misclassified by the FR model, which may be most informative for training (Swayamdipta et al., 2020). In contrast, FVSG ensures all demographic groups are well represented by directly optimizing the distribution of generated identities. Furthermore, FVSG is less dependent on the FR model because the model is used as a guidance signal without thresholding and does not risk removing informative cases.

| FVSG | S1F | LFW | CFP-FP | CPLFW | AgeDB | CALFW | Average | African | Asian | Caucasian | Indian | Average |
|---|---|---|---|---|---|---|---|---|---|---|---|---|
| ✓ | Q | **0.9938** | **0.9460** | 0.8882 | **0.9438** | 0.9305 | **0.9405** | **0.8895** | 0.8733 | **0.9295** | **0.8988** | **0.8978** |
| ✗ | Q | 0.9920 | 0.9433 | **0.8917** | 0.9412 | **0.9313** | 0.9399 | 0.8783 | **0.8780** | 0.9260 | 0.8917 | 0.8935 |
| ✗ | Q+ID | 0.9923 | 0.9451 | 0.8897 | 0.9388 | 0.9303 | 0.9393 | 0.8850 | 0.8695 | 0.9233 | 0.8928 | 0.8927 |

Table 7: **FVSG ablation study.** FVSG=Face Vendi Score Guidance. S1F=Stage 1 filtering. Q=quality. ID=identity. All datasets contain 0.5M images (50 images/ID).

**Filtering**. Filtering is widely used in synthetic face generation Kim et al. (2023); Wu et al. (2024a), although the importance of filtering has not been well studied. Similar to Wu et al. (2024a), we noticed that filtering only affected a small proportion of the generated images, and here we further investigated the impact of filtering on face verification accuracy (Table 8). While the best overall performance is observed with filtering, we observe that filtering is not necessary for VariFace to achieve state-of-the-art results. Even without any filtering, VariFace still considerably outperforms previous SOTA methods (0.9386 > 0.9200), highlighting the robustness of the conditional generation models used in VariFace.

| S1F | S2F | LFW | CFP-FP | CPLFW | AgeDB | CALFW | Average | African | Asian | Caucasian | Indian | Average |
|---|---|---|---|---|---|---|---|---|---|---|---|---|
| ✓ | ✓ | **0.9938** | 0.9460 | 0.8882 | **0.9438** | 0.9305 | 0.9405 | **0.8895** | 0.8733 | **0.9295** | **0.8988** | **0.8978** |
| ✓ | ✗ | 0.9937 | 0.9410 | 0.8892 | 0.9382 | 0.9308 | 0.9386 | 0.8867 | **0.8785** | 0.9247 | 0.8962 | 0.8965 |
| ✗ | ✓ | 0.9933 | 0.9444 | **0.8923** | 0.9413 | **0.9325** | **0.9408** | 0.8880 | 0.8722 | 0.9272 | 0.8918 | 0.8948 |
| ✗ | ✗ | 0.9908 | **0.9464** | 0.8912 | 0.9353 | 0.9293 | 0.9386 | 0.8828 | 0.8713 | 0.9212 | 0.8912 | 0.8916 |

Table 8: **Filtering ablation study.** SB=Standard Benchmark. S1F=Stage 1 filtering. S2F=Stage 2 filtering. All datasets contain 0.5M images (50 images/ID). The bottom row corresponds to results where no filtering was applied.

# 5  Conclusions

In this paper, we propose VariFace, a two-stage, diffusion-based pipeline for generating fair and diverse synthetic face datasets for training FR models. We introduce Face Recognition Consistency to refine demographic labels, Face Vendi Score Guidance to improve interclass diversity, and Divergence Score Conditioning to balance the ID preservation-intraclass diversity trade-off. When controlling for the dataset size and the number of images per ID, VariFace consistently outperforms previous synthetic dataset methods across six evaluation datasets and achieves FR performance comparable to real data (Table 1, 2). Furthermore, by scaling synthetic dataset size, VariFace outperforms the real dataset, as well as previous synthetic methods at all dataset sizes (Table 3, 4). Importantly, VariFace addresses fairness concerns, achieving better representation of minority demographic classes demonstrated both through qualitative visualizations (Fig. 4) and quantitative evaluation (Table 2).

With a gap between the FR performances from training on synthetic and real datasets, previous methods have emphasized the benefit of using synthetic data to augment real datasets (Qiu et al., 2021; Bae et al., 2023; Kim et al., 2024). In contrast, we demonstrate for the first time that state-of-the-art FR performance can be achieved when training only on synthetic data. Therefore, our results establish synthetic face datasets as a viable solution for achieving accurate and fair FR performance.

**Broader Impact Statement**

Over the past decade, there has been significant improvements in automated face recognition performance, with models achieving human-level performance (Taigman et al., 2014; Phillips et al., 2018). The performance improvement is not only due to architectural developments, but perhaps equally, if not more importantly, due to the significant increase in data used to train these models. However, these massive face datasets were obtained by web-scraping and do not have consent from individuals to use their faces for this purpose. To avoid the need to use these datasets, there has been growing interest to develop synthetic face datasets as an alternative. However, the best current synthetic methods, including our proposed method, still rely on real data to train the synthetic models. While this is a better option than directly training on real data, synthetic methods based on CG data or small-scale consented real face data are promising research directions that could eliminate the reliance on massive web-scraped datasets.

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

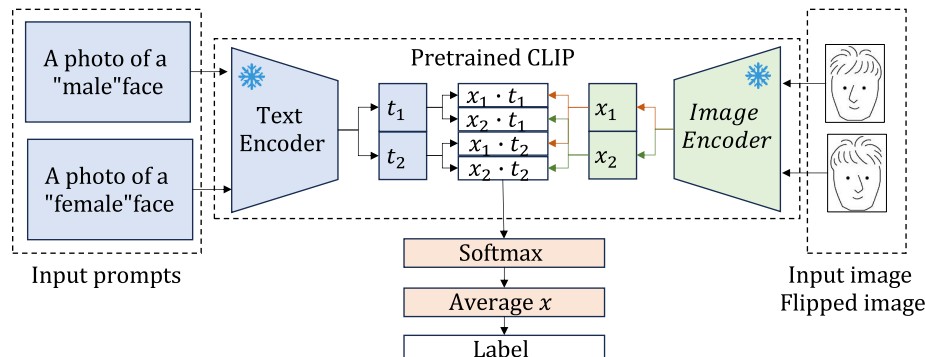

Figure S1: **CLIP demographic labeling.** Using a pretrained pair of CLIP image and text encoders, the cosine similarities between the image and text embeddings are computed and then converted into softmax probabilities. The final label is obtained after averaging softmax probabilities across values obtained from the image and flipped image embeddings.

## A    CLIP Demographic labeling

### A.1    Framework

We adapt the CLIP-IQA framework (Wang et al., 2023) to obtain demographic labels without relying on supervised models (Fig. S1). Unlike CLIP-IQA, we input both the image and its horizontally flipped version into the CLIP image encoder to obtain their corresponding image embeddings. We use the horizontally flipped image as a form of test-time augmentation to improve the robustness of the predictions. For the text encoder, we pass a set of prompts with the structure "A photo of a * face" where * is replaced with ["Male", "Female"], ["Young", "Old"], ["Caucasian", "Asian", "Indian", "African"] for gender, age and race labeling, respectively. Given a set of text embeddings $T = \{t_1, t_2, ..., t_n\}$ and the embedding of an image and its horizontally flipped version $X = \{x_1, x_2\}$, first the cosine similarity is computed:

$$s_{i,j} = \frac{x_j \cdot t_i}{\|x_j\|\|t_i\|}, i \in \{1, 2, ..., n\}, j \in \{1, 2\}. \tag{6}$$

Next, the softmax values, $\bar{s}_i$, for each label are computed and averaged over results from the image and flipped version:

$$\bar{s}_i = \frac{1}{|X|} \sum_j \frac{e^{s_{i,j}}}{\sum_i e^{s_{i,j}}}. \tag{7}$$

Finally, the value for the label, l, is obtained:

$$l = \arg\max_i \bar{s}_i. \tag{8}$$

### A.2    Evaluation

To evaluate the benefit of using CLIP compared to supervised models, we compare CLIP with DeepFace (Serengil & Ozpinar, 2021), a library that contains supervised models for race prediction of face images. Furthermore, we evaluate the benefit of including Face Recognition Consistency (FRC) with CLIP predictions.

For evaluation, we use the RFW dataset, which contains 48,000 images evenly divided across the following races: ["Caucasian", "Asian", "Indian", "African"]. DeepFace includes "Latino Hispanic" and "Middle Eastern" as predicted classes, and we relabel these as "Indian" predictions to conform to the RFW race

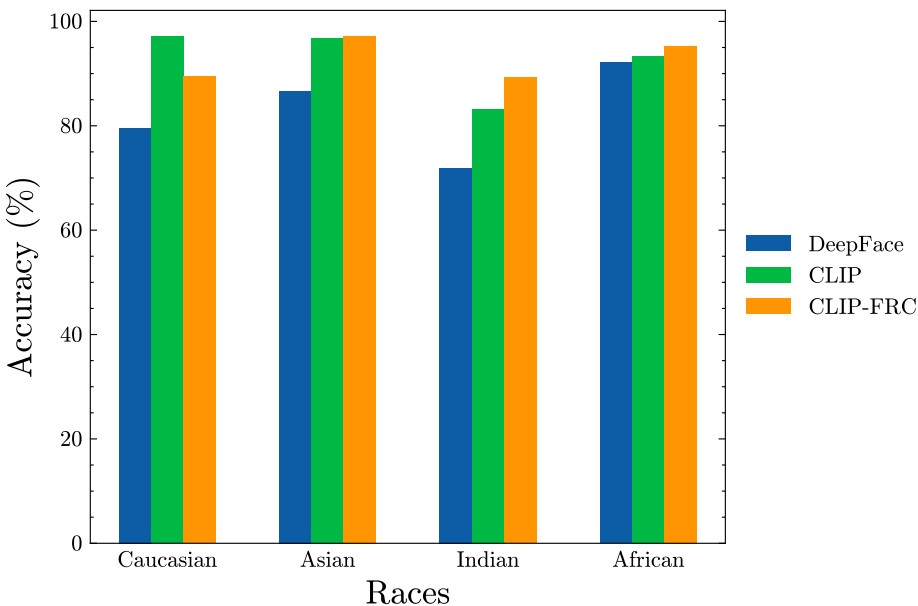

Figure S2: **Race prediction accuracy on the RFW dataset.**

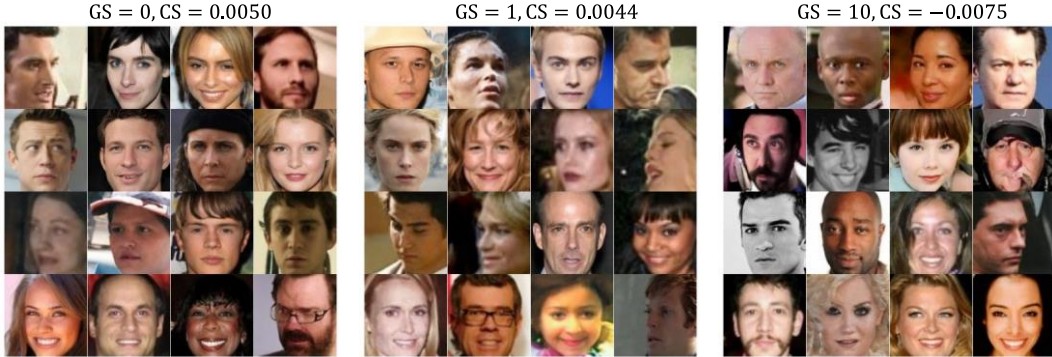

Figure S3: **Varying guidance scale with Face Vendi Score Guidance.** Examples were synthesized using VariFace. GS = guidance scale, CS = cosine similarity.

categories. For FRC, we set $K = 10$ to account for the small dataset size. The accuracy of the predictions using DeepFace, CLIP, and CLIP-FRC for each race are shown in Fig. S2.

CLIP outperforms DeepFace across all races, notably with respect to the Caucasian race prediction ($79.6\% \rightarrow 97.1\%$). Overall, the accuracy using DeepFace, CLIP, and CLIP-FRC are 82.6%, 92.6%, and 92.8%, respectively. Despite a decrease in performance on Caucasian individuals ($97.1\% \rightarrow 89.4\%$), there is an improvement across all other races with using FRC: Asian ($96.8\% \rightarrow 97.2\%$), Indian ($83.2\% \rightarrow 89.2\%$), and African ($93.3\% \rightarrow 95.3\%$).

## B   Face Vendi Score Guidance

To improve interclass diversity, we apply Face Vendi Score Guidance while sampling individuals in stage 1. The effect of varying the guidance scale is shown in Table S1, with examples shown in Fig. S3. At higher guidance scales, there is improved interclass diversity measured using cosine similarity. However, high guidance scales are associated with a higher frequency of artifacts, suggesting that careful selection of the guidance scale is required.

| Guidance scale | Cosine Similarity | High quality |
|:---:|:---:|:---:|
| 0 | 0.0032 | 0.62 |
| 1 | 0.0027 | 0.59 |
| 10 | 0.0025 | 0.44 |

Table S1: **Varying guidance scale with Face Vendi Score Guidance.** Cosine similarity is computed over an average of 10,000 embeddings. High quality refers to the fraction of images with a quality score > 0.7 evaluated by CLIB-FIQA.

| DS | LFW | CFP-FP | CPLFW | AgeDB | CALFW | Average | African | Asian | Caucasian | Indian | Average |
|:---:|:---:|:---:|:---:|:---:|:---:|:---:|:---:|:---:|:---:|:---:|:---:|
| $[0.5, 0.6]$ | 0.9900 | 0.9453 | 0.8872 | 0.9338 | 0.9227 | 0.9358 | 0.8772 | 0.8650 | 0.9113 | 0.8807 | 0.8835 |
| $[0.6, 0.7]$ | 0.9923 | 0.9443 | 0.8863 | 0.9405 | 0.9290 | 0.9385 | 0.8862 | 0.8720 | 0.9242 | 0.8878 | 0.8925 |
| $[0.7, 0.8]$ | 0.9918 | 0.9279 | 0.8747 | 0.9377 | 0.9288 | 0.9322 | 0.8792 | 0.8698 | 0.9233 | 0.8873 | 0.8899 |
| $[0.8, 0.9]$ | 0.9905 | 0.8737 | 0.8187 | 0.9088 | 0.9235 | 0.9030 | 0.8433 | 0.8420 | 0.8965 | 0.8607 | 0.8606 |
| $[0.5, 0.8]$ | **0.9938** | **0.9460** | **0.8882** | **0.9438** | **0.9305** | **0.9405** | **0.8895** | **0.8733** | **0.9295** | **0.8988** | **0.8978** |

Table S2: **Divergence score hyperparameter evaluation.** For each setting, we apply the Divergence score (DS) with a uniform distribution $[A, B]$. For each dataset, the best performance is highlighted in bold, and the second-best performance is underlined.

## C  Divergence Score Conditioning

### C.1  Label noise detection with Divergence scores

Given the use of web scraping to obtain large-scale face datasets such as CASIA-WebFace, there are potential issues with mislabeling of identities. The divergence score (DS) can be used to identify mislabeled identities, where a low DS suggests label noise. For synthetic data, filtering low DS can be used to remove cases where identity is not preserved. Examples of cases with low DS within the SynFace dataset are shown in Fig. S4.

The examples highlight how DS can be used to identify obvious cases where individual images differ from other images assigned the same ID label.

### C.2  Hyperparameter evaluation

To demonstrate the effect of hyperparameter settings for the DS, we evaluate a broad range of DS conditioning values (Table S2).

There is considerably lower performance when using either a low range of DS values ($[0.5, 0.6]$) or a high range of DS values ($[0.8, 0.9]$), which are associated with loss of ID preservation and low intraclass diversity, respectively. The best performance was observed with a broad range of DS values ($[0.5, 0.8]$), demonstrating the usefulness of DS to balance ID preservation and intraclass diversity.

## D  Age Conditioning

In the second stage, we apply age conditioning to generate face images of different ages. The benefit of face recognition performance, especially on age-diverse datasets, was shown in Table 6, and here we show examples of face images generated with different age conditions in Fig. S5.

The examples demonstrate that VariFace can generate face images across a broad range of ages while maintaining the identity of the individual.

## E  Synthetic faces comparison with real faces

One privacy concern with using deep generative models is the potential for real face images to leak into the synthetic dataset. To compare the identities in the synthetic and real datasets, we apply a pretrained face recognition model (IResNet-100) to generate embeddings and use these embeddings to determine identity

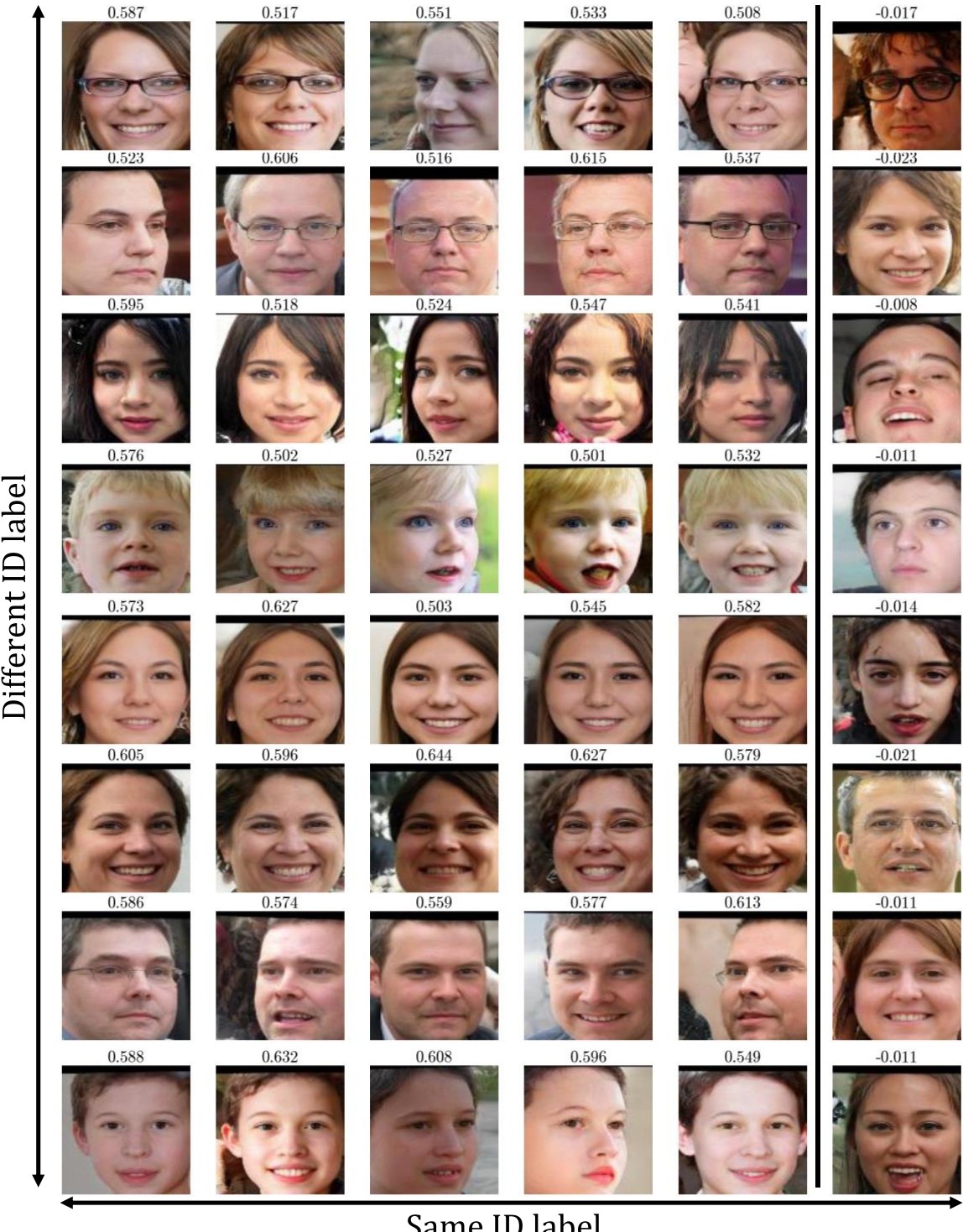

Figure S4: **Detecting label noise with Divergence Scores.** Example images with associated divergence scores for identities in the SynFace dataset. Cases with low divergence scores are shown in the column on the right.

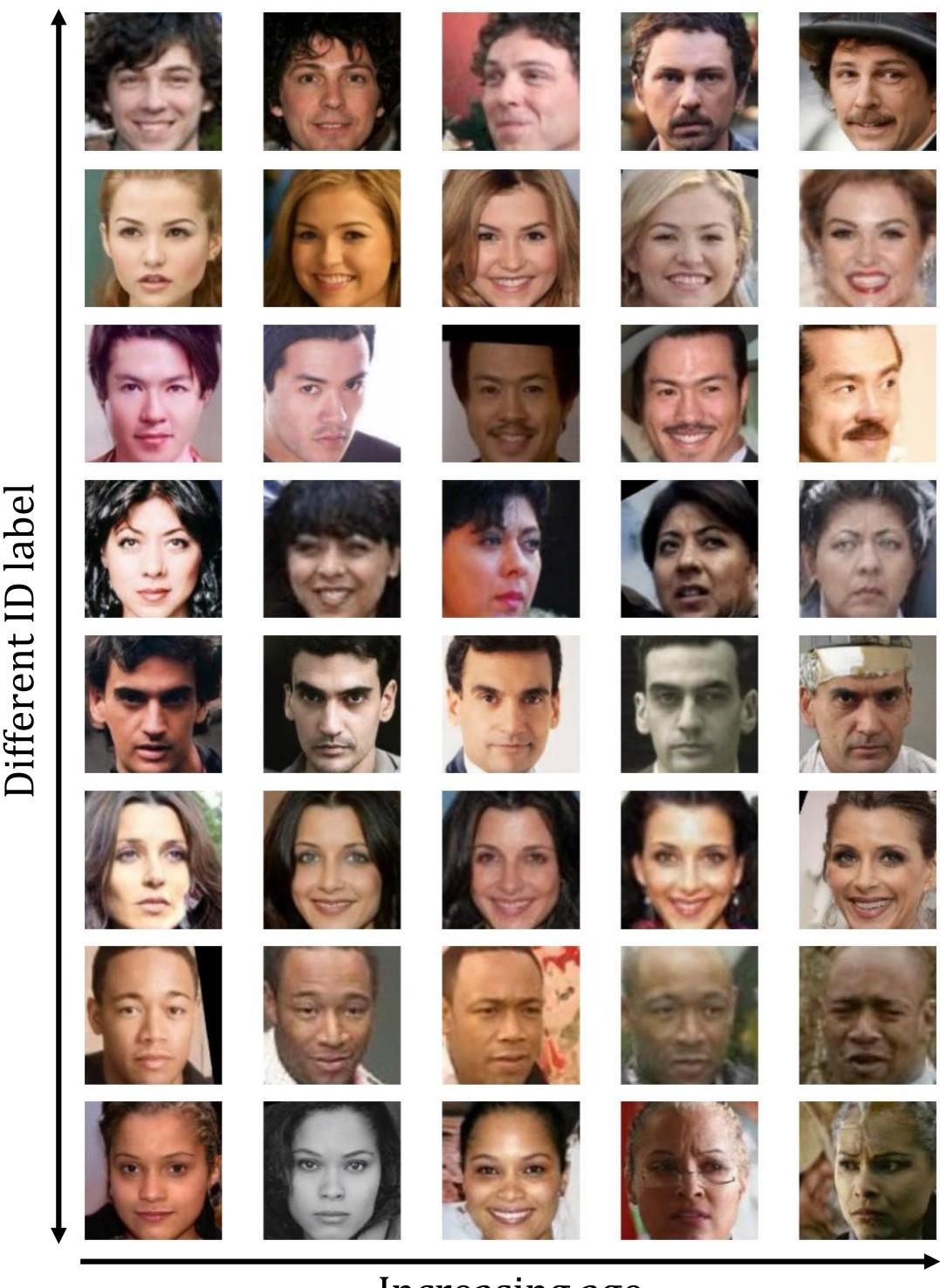

Figure S5: **VariFace can generate images of the same individual of different ages.** Example of synthetic images generated with different values for the age condition. For each row, the same ID condition was used, and the DS was fixed at 0.8 for all samples.

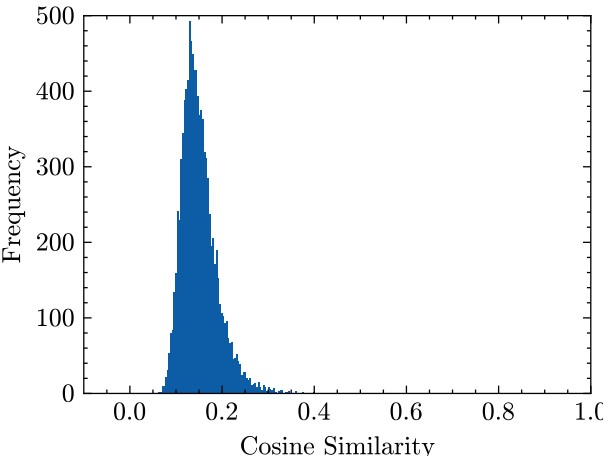

Figure S6: **Histogram of maximum cosine similarities of VariFace identities compared to CASIA-WebFace identities.**

similarity. For each of the 10,000 identities in the 0.5M VariFace dataset, we plot the maximum cosine similarity with the CASIA-WebFace dataset (Fig. S6).

Most synthesized identities have a low maximum cosine similarity score of around 0.2, with very few cases above 0.3. This suggests that most of the identities synthesized are different to those in the CASIA-WebFace dataset.

## F  Diffusion model settings

The settings for the diffusion models used in VariFace are shown in Table S3. We generally follow the default settings from the original HDiT, except for using Adaptive Discriminator Augmentation. (Karras et al., 2020), and modifying the conditioning module to accommodate for multiple conditions (Fig. S7).

## G  Face recognition model settings

The settings used for the FR model are shown in Table S4. We generally follow default settings from the InsightFace library (Deng et al., 2019), except for substituting the polynomial learning rate scheduler for a step learning rate scheduler and including additional data augmentations.

## H  Baseline real dataset performance

Due to differences in models, loss function, and training setup used, there is considerable variation in reported baseline FR performance obtained with CASIA-WebFace. To validate whether VariFace outperforms other baselines used in previous research, we compare our real dataset performance with others reported in the literature. The reported face recognition model performance trained on CASIA-WebFace and evaluated on the Standard Benchmark is shown in Table S5.

Our CASIA-WebFace trained FR model achieves performance comparable to the highest reported baselines. Importantly, even when compared to the best-performing baseline, our proposed method, VariFace, achieves better performance at 1.2M images $(0.9482 \rightarrow 0.9492)$.

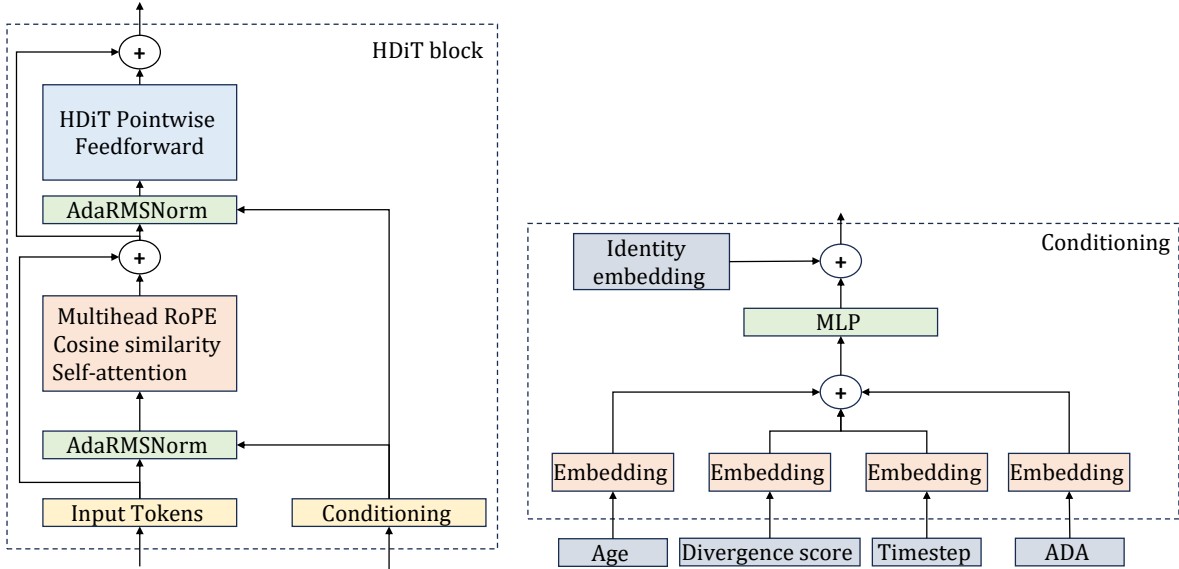

Figure S7: **VariFace HDiT block.** We follow the structure of the HDiT block used in (Crowson et al., 2024) and modify the conditioning block to handle multiple conditions. The structure of the conditioning block is shown for the stage 2 model. For the stage 1 model, the identity condition is removed, and age and divergence scores are replaced with race and gender labels. GEGLU is used as the MLP (Shazeer, 2020).

| Hyperparameter | Setting |
|---|---|
| Training steps | 700K (stage 1)/3M (stage 2) |
| Batch size | 128 |
| Hardware | 4 A100 |
| Training time | 19 hours/3.5 days |
| Patch size | 4 |
| Levels (Local + Global attention) | 1 + 1 |
| Depth | [2, 11] |
| Width | [256, 512] |
| Attention Head Dim | 64 |
| Neighborhood Kernel Size | 7 |
| Data Sigma | 0.5 |
| Sigma Range | [1e-3, 1e3] |
| Sigma Sampling Density | Interpolated cosine |
| Augmentation Probability | 0.12 |
| Dropout Rate | 0 |
| Conditional Dropout Rate | 0 |
| Optimizer | AdamW |
| Learning Rate | 0.0005 |
| Betas | [0.9, 0.95] |
| Eps | 1e-08 |
| Weight decay | 0.0001 |
| EMA Decay | 0.9999 |
| Sampler | DPM++(3M) SDE |
| Sampling steps | 50 |

Table S3: **Diffusion model settings.**

| Hyperparameter | Setting |
|---|---|
| Backbone | iResNet-50/iResNet-100 |
| Batch size | 256 |
| Epochs | 40 |
| Optimizer | SGD |
| Momentum | 0.9 |
| Weight decay | 5e-4 |
| Learning rate | 0.1 |
| Learning rate scheduler | Step LR (x0.1 at 24, 30, and 36) |
| Loss | ArcFace |
| Loss settings | scale=64, margin=0.5 |
| Augmentations | [transforms.ToPILImage(), transforms.RandomHorizontalFlip(), transforms.RandomAdjustSharpness(sharpness_factor=1.5, p=0.5), transforms.RandomAutocontrast(p=0.5), transforms.RandomEqualize(p=0.5), transforms.ToTensor(), transforms.Normalize(mean=[0.5, 0.5, 0.5], std=[0.5, 0.5, 0.5]), transforms.RandomErasing(p=0.5, scale=(0.02, 0.4)) ] |

Table S4: **Face recognition model settings.**

| Paper | LFW | CFP-FP | CPLFW | AgeDB | CALFW | Average |
|---|---|---|---|---|---|---|
| Kim et al. 2024 (Kim et al., 2024) | 0.9935 | 0.9597 | 0.8412 | 0.9365 | 0.9078 | 0.9277 |
| Kim et al. 2023 (Kim et al., 2023); Papantoniou et al. 2024 (Papantoniou et al., 2024) | 0.9942 | 0.9656 | 0.8973 | 0.9408 | 0.9332 | 0.9462 |
| Boutros et al. 2023 (Boutros et al., 2023b); Kolf et al. 2023 (Kolf et al., 2023) | **0.9955** | 0.9531 | 0.8995 | 0.9455 | 0.9378 | 0.9463 |
| Wu et al. 2024 (Wu et al., 2024a) | 0.9938 | **0.9691** | 0.8978 | 0.9450 | 0.9335 | 0.9479 |
| Boutros et al. 2023 (Boutros et al., 2023a) | 0.9952 | 0.9552 | **0.9038** | 0.9477 | **0.9393** | **0.9482** |
| Ours | 0.9950 | 0.9536 | 0.9007 | **0.9487** | 0.9368 | 0.9470 |

Table S5: **Face verification accuracy using CASIA-WebFace from previous research.** For each dataset, the best performance is highlighted in bold, and the second-best performance is underlined.

| Loss | LFW | CFP-FP | CPLFW | AgeDB | CALFW | Average | African | Asian | Caucasian | Indian | Average |
|---|---|---|---|---|---|---|---|---|---|---|---|
| ArcFace (Deng et al., 2019) | 0.9950 | **0.9536** | 0.9007 | 0.9487 | 0.9368 | **0.9470** | **0.8822** | **0.8697** | 0.9448 | **0.8978** | **0.8986** |
| CurricularFace (Huang et al., 2020) | 0.9920 | 0.9421 | 0.8813 | 0.9308 | 0.9247 | 0.9342 | 0.8463 | 0.8340 | 0.9163 | 0.8608 | 0.8644 |
| SphereFace (Liu et al., 2017) | 0.9943 | 0.9524 | **0.9018** | 0.9478 | **0.9380** | 0.9469 | 0.8787 | 0.8658 | 0.9415 | 0.9003 | 0.8966 |
| AdaFace (Kim et al., 2022b) | **0.9955** | 0.9534 | 0.9007 | 0.9485 | 0.9345 | 0.9465 | 0.8787 | 0.8648 | 0.9452 | 0.8950 | 0.8959 |
| MagFace (Meng et al., 2021) | 0.9940 | 0.9517 | 0.8952 | **0.9488** | 0.9332 | 0.9446 | 0.8800 | 0.8628 | **0.9453** | 0.8900 | 0.8945 |
| UniFace (Zhou et al., 2023) | 0.9945 | 0.9531 | 0.9008 | 0.9437 | 0.9348 | 0.9454 | 0.8712 | 0.8583 | 0.9437 | 0.8917 | 0.8912 |

Table S6: **Face verification accuracy using CASIA-WebFace with different loss functions.** For each dataset, the best performance is highlighted in bold, and the second-best performance is underlined.

| Dataset | Size | LFW | CFP-FP | CPLFW | AgeDB | CALFW | Average | Real Gap | Orig Gap |
|---|---|---|---|---|---|---|---|---|---|
| CASIA-WebFace | 0.5M | 0.9950 | 0.9536 | 0.9007 | 0.9487 | 0.9368 | 0.9470 | 0.0000 | - |
| SynFace (Qiu et al., 2021) | 0.5M | 0.8580 | 0.6473 | 0.6395 | 0.5765 | 0.6987 | 0.6840 | -0.2630 | - |
| SynFace (Qiu et al., 2021) | 1.0M | 0.8593 | 0.6341 | 0.6452 | 0.5750 | 0.6860 | 0.6799 | -0.2671 | - |
| DigiFace (Bae et al., 2023) | 0.5M | 0.8508 | 0.7431 | 0.6498 | 0.6093 | 0.6713 | 0.7049 | -0.2421 | -0.1296 |
| DigiFace (Bae et al., 2023) | 1.2M | 0.8872 | 0.7827 | 0.6835 | 0.6218 | 0.7170 | 0.7384 | -0.2086 | -0.1148 |
| DCFace (Kim et al., 2023) | 0.5M | 0.9863 | 0.8896 | 0.8325 | 0.9082 | 0.9173 | 0.9068 | -0.0402 | +0.0112 |
| DCFace (Kim et al., 2023) | 1.2M | 0.9895 | 0.886 | 0.8497 | 0.9173 | 0.9268 | 0.9139 | -0.0331 | +0.0018 |
| Vec2Face (Wu et al., 2024a) | 0.5M | 0.9848 | 0.8737 | 0.8413 | 0.9187 | 0.9298 | 0.9097 | -0.0373 | -0.0103 |
| Vec2Face (Wu et al., 2024a) | 1.0M | 0.9868 | 0.8819 | 0.8537 | 0.9408 | 0.9372 | 0.9201 | -0.0269 | -0.0046 |

Table S7: **Face Verification Accuracy using open-source synthetic datasets.** The Real Gap is the difference to the real dataset performance. The Orig Gap is the difference to the performance reported in the original paper.

## I  Effect of loss function on model performance

Since the development of the ArcFace loss, there have been numerous alternative loss functions proposed in the literature (Liu et al., 2017; Huang et al., 2020; Meng et al., 2021; Kim et al., 2022b; Zhou et al., 2023). The performance using different loss functions for FR models trained on the CASIA-WebFace dataset is shown in Table S6. Generally, the performance is similar across loss functions, with the best performance obtained using the ArcFace loss.

## J  Reproducing performance from open-source synthetic datasets

To further compare our method with current state-of-the-art synthetic datasets, we train FR models using the same settings as our experiments on open-source synthetic face datasets. The results are shown in Table S7.

Except for DigiFace, we observed a similar performance with our training settings and the original paper results. Our performance obtained with DigiFace is significantly lower than the results in the original paper, and this is likely due to differences in the data augmentation strategies used (Bae et al., 2023). While our data augmentation settings are tuned for real data, DigiFace employs additional data augmentation to overcome the domain gap between real and CG data. Importantly, regardless of whether the original dataset performance or our reproduced performances are used, our proposed method remains the best-performing method.

## K  Additional fairness comparison: Skin color bias

Fairness is a broad concept that encompasses a variety of ethical and legal principles to ensure equitable treatment of individual and groups across different contexts. While we focus on race and gender due to their legal significance and availability of evaluation datasets such as RFW (Wang et al., 2019), race labels are not comprehensive and do not account for the diversity within, and overlap between, each race category. While not as legally significant, skin color represents faces on a continuous and comprehensive scale, providing a complementary perspective to evaluate the fairness of real and synthetic face datasets. Here, we analyze skin color bias, adopting the multidimensional skin color scale in Thong et al. (2023), which defines apparent skin color in terms of luminance $L^*$ and hue angle $h^*$, with:

$$h^* = \arctan(b^*/a^*), \tag{9}$$

where $L^*$, $a^*$ and $b^*$ refer to the CIELAB color space. With these two metrics, the skin color can be categorized into light ($L^* > 60$) or dark ($L^* \leq 60$) skin tones, as well as red ($h^* \leq 55$) and yellow ($h^* > 60$) skin hues. We use DeeplabV3 (Chen et al., 2018) to extract face segmentation masks and compare 10,000 sampled individuals from CASIA-WebFace, VariFace and DCFace. The results are shown in Table S8.

|     |        | Skin Tone | | |
|-----|--------|-------|------|-------|
|     |        | Light | Dark | Total |
| Hue | Red    | **44.13** | 28.26 | 72.39 |
|     | Yellow | 17.49 | 10.12 | 27.61 |
|     | Total  | 61.62 | 38.38 | 100 |

(a) CASIA-WebFace

|     |        | Skin Tone | | |
|-----|--------|-------|------|-------|
|     |        | Light | Dark | Total |
| Hue | Red    | **44.14** | 31.46 | 75.60 |
|     | Yellow | 15.86 | 11.24 | 27.10 |
|     | Total  | 60.00 | 40.00 | 100 |

(b) DCFace

|     |        | Skin Tone | | |
|-----|--------|-------|------|-------|
|     |        | Light | Dark | Total |
| Hue | Red    | **33.13** | 30.61 | 63.74 |
|     | Yellow | 21.87 | 14.39 | 36.26 |
|     | Total  | 55.00 | 45.00 | 100 |

(c) VariFace

Table S8: **Skin color bias** in various face datasets (in %). VariFace reduces the skin color bias present in CASIA-WebFace.

In line with results from Thong et al. (2023), real face datasets such as CASIA-WebFace demonstrate considerable bias towards light-red skin tones, with low representation of dark-yellow skin tones. DCFace also demonstrates a similar degree of skin color bias. In contrast, VariFace reduces the skin color bias compared to CASIA-WebFace, with a considerable reduction in light-red representation (44.13 → 33.13) and improvement in dark-yellow representation (10.12 → 14.39). This demonstrates that VariFace not only reduces the bias with respect race labels, but generates a more diverse underlying distribution of individuals, including those reflecting underrepresented skin colors.

## L  Limitations

VariFace does not outperform the real dataset performance when using a fixed dataset size and images per ID. However, the ease of scalability is a key feature of synthetic methods, and VariFace outperforms the real dataset at larger dataset sizes. Another limitation is that by implicitly learning intraclass variation from the training dataset, the synthetic dataset diversity is necessarily limited by the diversity of the training dataset. Finally, the results presented here relied on training on CASIA-WebFace, a large-scale web-scraped dataset. With increasingly stricter regulations on collecting face datasets, small-scale consented face data as well as CG data provides alternatives to avoid using web-scraped data, but will require further work to address issues with limited diversity and domain gap.

## M    Additional VariFace Examples

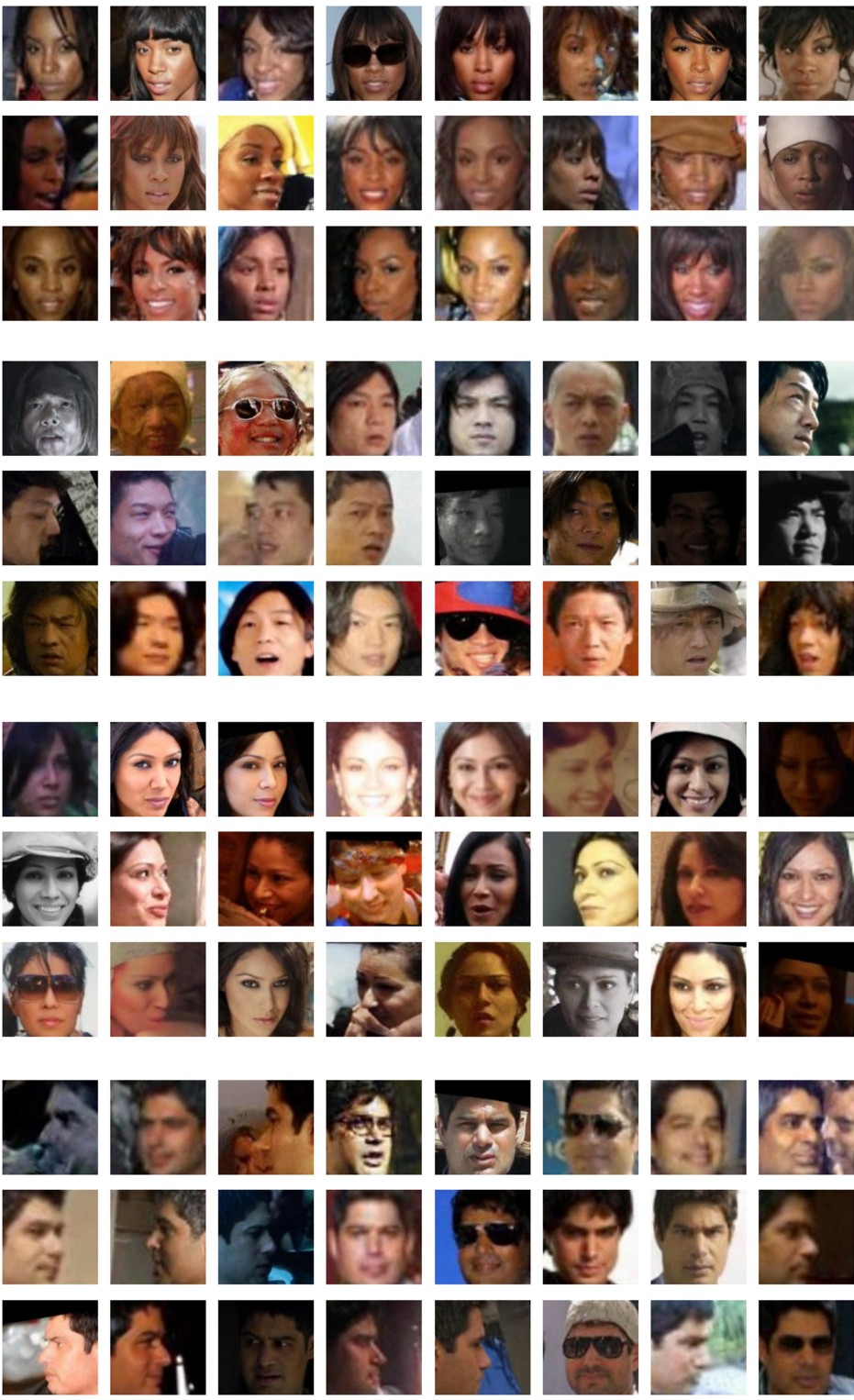

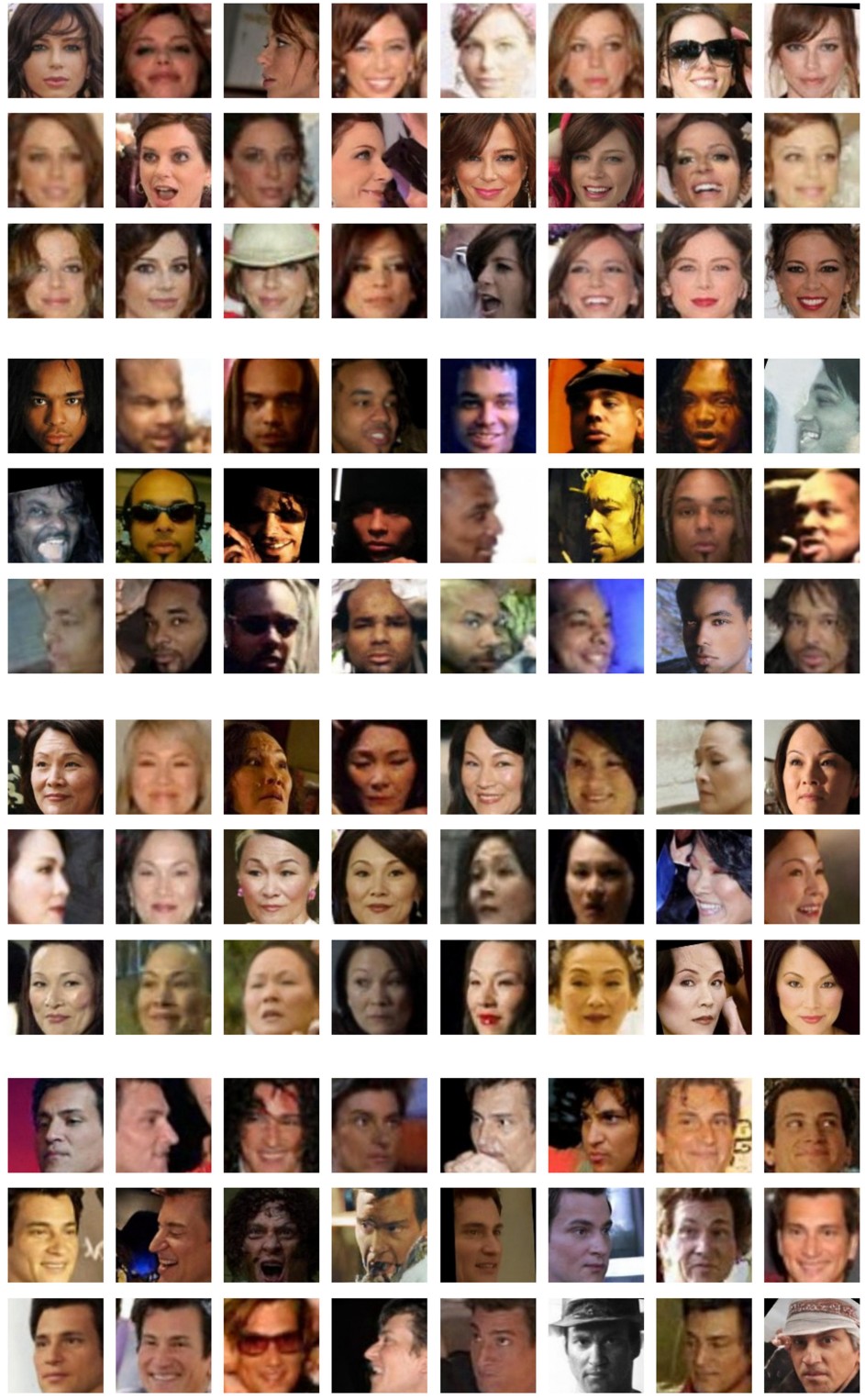

Figure S8: **VariFace generated examples**. Synthetic images generated with randomly sampled Divergence scores (0.5-0.8) without filtering.

