# OpenReview forum: "VariFace: Fair and Diverse Synthetic Dataset Generation for Face Recognition"
_TMLR — Rejected by TMLR_

### Review · Reviewer_1j7j · 2025-05-17

**Summary Of Contributions:**

This paper addresses an important problem: how to deal with the privacy risks and biases of the dataset used for training face recognition deep learning models.
Although there are many works tried to deal with this issue previously using synthetic dataset, those synthetic datasets support limited geographical diversities.
This paper proposes a diffusion-model based method that addresses the above issues and achieve high face recognition performance.

**Audience:**

Yes

**Broader Impact Concerns:**

Overall I like the direction the authors focus on and the issue they want to address.
It is undoubtly an important issue to work on.
There are some unclear point that makes me confuse about how the method really achieve the claimed results.
Meanwhile, I also not sure about how practically useful the proposed method is.
That is, given this method, if we can say that synthetic dataset is likely or has a chance to surpass real dataset, will any existing large public models or models from big company switch to use it?

**Claims And Evidence:**

Yes

**Requested Changes:**

I have listed the questions and changes I want to see in the revision in the "Strengths and Weaknesses" section.
I look forward authors feedback and revision.

**Strengths And Weaknesses:**

**Strengths**
- the dataset generated by this method achieve a new state-of-the-art FR performance across six face evaluation datasets using only synthetic data.
- The method is well designed and aims for generate interclass and intraclass balanced dataset.


**Weaknesses**
- I think the claim in abstract is a bit overclaim: "In an unconstrained setting, VariFace outperform real dataset". I personally think if this unconstrained setting means the dataset sizes are not the same between VariFace and the real dataset, then it is meaningless. Since that means the synthetic method (VariFace) can unlimitedly increase the size of the dataset. As discussed in Sec. 4.4, I think the synthetic dataset size is 5x than the real world dataset. I might be wrong here, but I would love authors to discuss this matter more.

- Although the goal is to get a balanced and unbiased dataset, I am unsure why using CLIP can help this since I think there is also biased in the pretrained CLIP model.

- There are some descriptions in the paper are not clear and vague when I read it;
   - Sec. 3.1 -> "we first obtain initial estimates.." -> what is "initial estimates"?
   - Sec. 3.1 -> "let I represent an image from the face dataset D" -> is this dataset a synthetic dataset or sample an image from a real image dataset? And when CLIP is used? Since in this part, the authors keep mentioning "using a pretrained FR model". Which FR model is used?
   - Sec. 3.2 -> "The second stage creates diverse image variations for each face generated in the first stage while preserving
face ID." -> why? I am unsure why this is the goal for "Diverse intraclass variation"?

- Overall, I am quite confuse that, the proposed method rely on one FR model which is trained on real world face dataset. That means, there is biased in the FR model. And if the proposed method rely on it, will the biased be inherited? I would love to see authors discuss more about this matter.

---

> ### Author Response · Authors · 2025-06-06
> **Reply to comments from Reviewer 1j7j**
>
> (Due to 5000 character limit, we will split our response into multiple comments).
>
> We thank the reviewer for their insightful comments and we are glad to hear that the reviewer recognizes the importance of the topic and the merits of our approach. We have updated the manuscript accordingly and here we would like to address each of the reviewer’s concerns and provide some additional clarifications:
>
> **I think the claim in abstract is a bit overclaim: "In an unconstrained setting, VariFace outperform real dataset". I personally think if this unconstrained setting means the dataset sizes are not the same between VariFace and the real dataset, then it is meaningless. Since that means the synthetic method (VariFace) can unlimitedly increase the size of the dataset. As discussed in Sec. 4.4, I think the synthetic dataset size is 5x than the real world dataset. I might be wrong here, but I would love authors to discuss this matter more.** \
> We believe that outperforming the real dataset even in an unconstrained setting is a significant milestone to reach. Prior to our method, no proposed synthetic face generation method, scaled to any dataset size (the largest being 15M images with Vec2Face), has outperformed models trained on real (CASIA-WebFace) data. We highlight in Figure 1 that performance inevitably plateaus when synthetic data is scaled, with all methods besides VariFace plateauing below the original dataset performance. We hypothesize that this saturating performance is a result of a lack of intraclass and interclass diversity, in addition to exacerbating the dataset bias. In contrast, VariFace mitigates the dataset bias (CLIP-FRC), and facilitates generation of intraclass (Divergence Score Conditioning) and interclass (Face Vendi Score Guidance) variation, enabling VariFace to outperform the real dataset when scaled. We have modified the manuscript to include this clarification (Introduction).
>
> **Although the goal is to get a balanced and unbiased dataset, I am unsure why using CLIP can help this since I think there is also biased in the pretrained CLIP model.** \
> We agree with the reviewer and provide evidence that CLIP displays bias (Figure S2). However, previous methods rely on supervised models such as DeepFace to label races, and in the same figure we show that, compared to supervised methods (DeepFace), CLIP is less biased and has better accuracy across races. Moreover, to further mitigate CLIP bias, we propose CLIP-FRC, which applies a consistency constraint with face recognition model embeddings to refine CLIP-generated demographic labels.
>
> **Sec. 3.1 -> "we first obtain initial estimates.." -> what is "initial estimates"?** \
> The initial estimates refers to race and gender predictions generated by CLIP (we describe our method to generate these predictions in Section A of the supplementary materials). We call these the “initial” estimates because we propose refining these predictions using Face Recognition Consistency (FRC) to create the final predictions for race and gender. We have modified the manuscript to improve clarity (Section 3.1).
>
> **Sec. 3.1 -> "let I represent an image from the face dataset D" -> is this dataset a synthetic dataset or sample an image from a real image dataset?** \
> The face dataset D refers to the real dataset used for training, which in our experiments is CASIA-WebFace. We have added this as clarification in the manuscript (Section 3.1).
>
> **And when CLIP is used?** \
> CLIP is used in Stage 1 of the pipeline, to generate initial race and gender prediction labels that are subsequently refined by FRC to generate the final labels used to train the Stage 1 diffusion model (Figure 2).
>
> **Since in this part, the authors keep mentioning "using a pretrained FR model". Which FR model is used?** \
> The pretrained model we use is an IResNet-100 that we train on CASIA-WebFace (Sec 4.1).
>
> **Sec. 3.2 -> "The second stage creates diverse image variations for each face generated in the first stage while preserving face ID." -> why? I am unsure why this is the goal for "Diverse intraclass variation"?** \
> Intraclass variation refers to the diversity of images for a given face identity, where diversity could include differences in face pose, lighting, expression etc. The second stage of our pipeline is responsible for generating the diversity for a given face identity. Importantly, there is a trade-off between diversity and maintaining the identity of the individual (for example, more extreme lighting conditions adds diversity but makes it harder to identify a person), and optimizing this trade-off without supervised labels is one of our key contributions (Divergence Score Conditioning).

---

> > ### Author Response · Authors · 2025-06-06
> > **Additional reply to comments from Reviewer 1j7j**
> >
> > **Overall, I am quite confuse that, the proposed method rely on one FR model which is trained on real world face dataset. That means, there is biased in the FR model. And if the proposed method rely on it, will the biased be inherited? I would love to see authors discuss more about this matter.** \
> > While it is true that pretrained models will inherit biases from the training data, we use two strategies to mitigate the bias. To generate fair and diverse interclass variation, we condition the stage 1 model on race and gender labels and apply Face Vendi Score Guidance, respectively. This ensures that the synthetic dataset contains a balanced set of identities across race and genders, mitigating the imbalance present in the training data. Secondly, we apply Divergence Score Conditioning that ensures all synthetic identities contain diverse intraclass variation. Therefore, by explicitly controlling both the interclass and intraclass variation, we are able to minimize biases inherited by the pretrained FR model.

---

### Review · Reviewer_YoMa · 2025-05-19

**Summary Of Contributions:**

This paper propose VariFace to generate a synthetic face recognition dataset for addressing privacy concern and race/gender balance. This method consider two-stage training and generation pipeline which is not commonly conisdered in SFR area.

Face Recognition Consistency, Face Vendi Score Guidance and Divergence Score Conditioning are proposed to improve the generation quality in terms of the diversity and ID-preserving.

**Audience:**

Yes

**Claims And Evidence:**

Yes

**Requested Changes:**

Please see the Strengths And Weaknesses

**Strengths And Weaknesses:**

Strengths:
1. This paper proposes to control race and gender distribution which is a important issue in face recognition dataset.

2. Results on the Synthetic Face Recognition evaluation dataset are good, exceeding the general real FR dataset CASIA-WebFace.

3. Ablation study on the condition proves the effectiveness of the chosen condition


Weaknesses:

1.Some key references are missing, SynthDistill (IJCB 2023) [1], CemiFace (NeurIPS 2024) [2] and ID^{3} (NeurIPS 2024) [3].

2. This paper proposes two-stage generation and filtering; however, the reference [1] has adopted a two-stage GAN method. Additionally, [1] has achieved dramatically better performance than the general one-stage (without filtering) method. A comparison with this paper in terms of method and experiment is needed.

3. This paper proposes to mitigate the privacy problem. However, at each stage, the pertained FR model is involved, which is derivative from an illegal dataset(like CASIA). For example, Stage 1: CLIP/ pertained FR, and stage 2: Face Vendi Score Guidance also involve retrained FR.

4. Figure 2, "a pretrained FR model is used to refine race (R) and gender (G) labels”, how to do it?

5 .Introducing Vendi Score Guidance is interesting, butit  is not novel, as the original idea is from Universal Guidance

7. Two-stage training and filtering introduce too much computational cost. Can you specify how long each stage takes? Are the diffusion models pretrained?

8. Does the second stage improve the overall performance? How about involving more filtering stages? And experiment with adding all the components in stage 1?

9. This paper proposes the Diversity score, which is the similarity to the average identity center. Do the authors consider the learned identity center in the pretrained FR linear layer? This idea is not new, as it is discussed in the missing reference [2].

10. Discussion between the proposed method with ID^3 [3] is needed as it uses a few pretrained models to produce generalised performance.

11. IResNet-100 and IResNet-50 are both adopted. More explanation is needed to demonstrate this choice.

12. For the experiment, please provide fair data volume for comparison, for example, DCFace use 20 K * 50, however, this paper only reports 50k*20.

Reference:

[1]Shahreza, Hatef Otroshi, Anjith George, and Sébastien Marcel. "Synthdistill: Face recognition with knowledge distillation from synthetic data." 2023 IEEE International Joint Conference on Biometrics (IJCB). IEEE, 2023.

[2]Sun, Zhonglin, et al. "CemiFace: Center-based Semi-hard Synthetic Face Generation for Face Recognition." The Thirty-eighth Annual Conference on Neural Information Processing Systems.

[3]Xu, Jianqing, et al. "$\text {ID}^ 3$: Identity-Preserving-yet-Diversified Diffusion Models for Synthetic Face Recognition." Advances in Neural Information Processing Systems 37 (2024): 77777-77798.

---

> ### Author Response · Authors · 2025-06-06
> **Reply to comments from Reviewer YoMa**
>
> (Due to 5000 character limit, we will split our response into multiple comments).
>
> We appreciate the reviewer for their detailed and useful comments. We are encouraged to hear that the reviewer recognizes the importance of the topic and effectiveness of our approach. We have updated the manuscript based on the comments and here we would like to address each of the reviewer’s concerns and provide some additional clarifications:
>
> **1. Some key references are missing, SynthDistill (IJCB 2023) [1], CemiFace (NeurIPS 2024) [2] and ID^{3} (NeurIPS 2024) [3].** \
> We thank the reviewer for the additional references. We have included these methods in the related works section and added CemiFace to our comparisons (Figure 1, Table 1, and Table 3). We note that ID^3 was originally included in the comparisons (Table 1). We do not include ID^3 in the unconstrained evaluation (Table 3) because only results for 0.5M dataset size were presented in the original paper. We include SynthDistill in the related works but not in the quantitative comparisons and the reason for this is provided below.
>
> **2. This paper proposes two-stage generation and filtering; however, the reference [1] has adopted a two-stage GAN method. Additionally, [1] has achieved dramatically better performance than the general one-stage (without filtering) method. A comparison with this paper in terms of method and experiment is needed.** \
> While the results presented in SynthDistill are impressive, we did not include a comparison to SynthDistill because the results presented in the SynthDistill paper are not comparable with our method. The main reason is that SynthDistill uses a teacher model trained on the MS-Celeb dataset (now retracted), while we benchmark our method against those trained on the CASIA-WebFace dataset. The MS-Celeb dataset contains 10M images, which is more than 20x the size of CASIA-WebFace. Moreover, SynthDistill uses an EfficientNet variant (TinyNet) rather than ResNet-50 as the FR model, with 1M+ samples (while in our constrained setting we benchmark 500K images with 50 IDs/image). Finally, the design of SynthDistill involves an online dataset generation process, while our method and the previous used in our benchmarks generate a fixed training dataset.
>
> **3. This paper proposes to mitigate the privacy problem. However, at each stage, the pertained FR model is involved, which is derivative from an illegal dataset(like CASIA). For example, Stage 1: CLIP/ pertained FR, and stage 2: Face Vendi Score Guidance also involve retrained FR.** \
> As acknowledged in the paper, we recognize the concerns around the use of web-scraped datasets and do not endorse their use. In this work, we use CASIA-WebFace solely to enable a fair comparison with prior methods. However, our approach is general and can readily be applied to datasets collected with proper consent. Notably, the pretrained model used in our pipeline is trained exclusively on the same dataset as the generative models, ensuring that our method relies only on the specified training data. This is in contrast to other methods, which supplement CASIA-WebFace with models trained on additional datasets such as FFHQ (DCFace), Glint360K (Vec2Face), and WebFace-4M (CemiFace).
>
> **4. Figure 2, "a pretrained FR model is used to refine race (R) and gender (G) labels”, how to do it?** \
> We propose the “Face Recognition Consistency” method to refine the race and gender labels, and provide the details in Sec 3.1. To summarize FRC, we leverage the finding that the embedding space of pretrained FR models are structured with respect to demographic information by using a clustering approach that enforces that identities close in FR embedding space should belong to the same demographic group.
>
> **5. Introducing Vendi Score Guidance is interesting, butit is not novel, as the original idea is from Universal Guidance** \
> We acknowledge in the paper that Face Vendi Score Guidance is similar to Universal Guidance, but we also believe there are significant differences. The motivation behind Universal Guidance is for controllable generation, where inputs such as segmentation masks and bounding boxes are used to guide the generation process. In contrast, we apply guidance in a novel way to improve face diversity, where our guidance signal acts to diversify rather than constrain the outputs. Moreover, we show empirically that guidance provides a better mechanism to enforce diversity than ID filtering (Table 7), demonstrating this method is a viable alternative to the common approach used by prior methods.

---

> > ### Author Response · Authors · 2025-06-06
> > **Additional reply to comments from Reviewer YoMa**
> >
> > **7. Two-stage training and filtering introduce too much computational cost. Can you specify how long each stage takes? Are the diffusion models pretrained?** \
> > Using 4 NVIDIA A100 GPUs, the training for stage 1 and stage 2 diffusion models takes 19 hours and 3.5 days, for 700K and 3M iterations, respectively (Sec 4.1). The filtering process uses negligible computational cost and, as we show in our ablation studies (Table 8), can be removed entirely while still maintaining state-of-the-art performance. The diffusion models are trained from scratch to ensure that identity information is only derived from the training data, in contrast to other methods such as those that use Stable Diffusion (Arc2Face), where identities may leak from pretrained data. In Section 4.6, we show that we can generate face datasets significantly faster with VariFace than current state-of-the-art methods including other diffusion-based (DCFace) and GAN-based (Vec2Face) pipelines (note Vec2Face bottleneck is not the GAN inference but the AttrOP method). While diffusion models involve sacrificing speed for better diversity and higher quality images compared to GANs, we believe that in the context of face image generation, it is important to generate realistic and diverse face data in a reasonable amount of time which motivates our choice of diffusion over GAN-based models.
> >
> > **8. Does the second stage improve the overall performance? How about involving more filtering stages? And experiment with adding all the components in stage 1?** \
> > By design, VariFace requires both stages to generate a face dataset. This is because the first stage generates a balanced set of synthetic identities that is used as a conditioning signal into the second stage to generate a diverse set of face images for each synthetic identity. The output of the second stage is the synthetic face dataset generated by VariFace. Regarding adding more filtering stages, from our experiments, we observed that we are able to achieve state-of-the-art performance without any filtering (Table 8) but observe a small performance improvement with the addition of filtering. Therefore, while it may be possible to add additional filtering stages, we do not believe it is a necessary component of our pipeline.
> >
> > **9. This paper proposes the Diversity score, which is the similarity to the average identity center. Do the authors consider the learned identity center in the pretrained FR linear layer? This idea is not new, as it is discussed in the missing reference [2].** \
> > While both VariFace and CemiFace use similarity to identity centers as a condition, we believe there are significant differences in the application of these methods. VariFace uses the embedding rather than linear layer to compute similarities, and this is crucial for our design. This is because the linear layer restricts identities to those of the training dataset, while using the embedding layer provides flexibility to use identities beyond those of training data. Therefore, while CemiFace is restricted to using identities from CASIA-WebFace, VariFace is able to use synthetically generated identities (which we generate from stage 1) and create variation of these faces. Moreover, CemiFace assumes that the similarity condition provides sufficient age diversity, but we highlight this is not the case and that age conditioning, in addition to similarity conditioning, provides significant improvements to FR performance (Table 6). Finally, we note that empirically VariFace achieves better performance compared to CemiFace across the 5 evaluation datasets in both constrained and unconstrained settings (Tables 1 and 3).
> >
> > **10. Discussion between the proposed method with ID^3 [3] is needed as it uses a few pretrained models to produce generalized performance.** \
> > We thank the reviewer for this suggestion and have added ID^{3} in the related works. We note that ID^{3} achieves good FR performance but falls behind similar recent methods such as Vec2Face and Arc2Face, and our method is able to achieve significantly better performance. One key difference between our method and ID^{3} is that ID^{3} relies on pretrained models to encode facial attributes, while our Divergence Score Conditioning method is able to generate intraclass variation without explicit attributes. While this limits ID^{3} to modeling age and pose variation, our method is able to generate more diverse variation by learning diversity from the dataset (Figure S8 shows examples of diverse variation generated by VariFace).

---

> > > ### Author Response · Authors · 2025-06-06
> > > **Further additional reply to comments from Reviewer YoMa**
> > >
> > > **11. IResNet-100 and IResNet-50 are both adopted. More explanation is needed to demonstrate this choice.** \
> > > We adopted the IResNet-50 network for evaluation to ensure a fair comparison with previous methods (Table S5). Within VariFace, we chose the larger IResNet-100 as a pretrained FR model for use in generating divergence scores as well as filtering. This is because IResNet-100 shows better FR performance and therefore we hypothesize that this model can provide more robust embeddings than IResNet-50. There is negligible computational cost to using IResNet-100 compared to IResNet-50 in VariFace as the main bottleneck lies in the stage 2 diffusion process.
> > >
> > > **12. For the experiment, please provide fair data volume for comparison, for example, DCFace use 20 K * 50, however, this paper only reports 50k*20.** \
> > > For comparisons with previous methods, we endeavored to include two sets of results: the performance under constrained setting (500K images with 50 images per identity), as well as the best performance reported by the paper without any constraints reagrding the number of images or identities. For DCFace, we have included both settings, with Table 1 showing the performance under constrained settings (10K x 50) and unconstrained settings (20K x 50 + 40K x 5). The unconstrained setting result is the best performance described in the DCFace paper, using 1.2M images.

---

### Review · Reviewer_r5au · 2025-05-28

**Summary Of Contributions:**

This paper introduces VariFace, a two-stage diffusion-based pipeline for generating large-scale, fair, and diverse synthetic face datasets for training face recognition models. The work addresses privacy and bias concerns associated with real web-scraped datasets, while also tackling the limited diversity found in existing synthetic datasets. The authors propose three key methodological contributions: Face Recognition Consistency for refining demographic labels, Face Vendi Score Guidance for enhancing interclass diversity, and Divergence Score Conditioning for managing the trade-off between identity preservation and intraclass diversity.

**Audience:**

Yes

**Broader Impact Concerns:**

The Broader Impact Statement correctly identifies the problem of the historical reliance on large-scale web-scraped datasets without consent in face recognition. The direction of the proposed solution, although, relies on CASIA-WebFace for training the VariFace generator itself, meaning that the ethical concerns tied to large-scale web-scraping are not fully severed but rather are fundamental to the proposed method.

**Claims And Evidence:**

Yes

**Requested Changes:**

While the experimental results are satisfactory, the paper would benefit from a more comprehensive discussion in two key areas. First, the reliance on web-scraped data requires deeper analysis: although VariFace aims to reduce dependence on such data for FR model training, the generator itself still relies on CASIA-WebFace, which only partially addresses the underlying ethical concerns. Second, the evaluation of fairness needs expansion beyond the current demographic balance approach, potentially incorporating further axes. From a technical perspective, clarification is needed on the Vendi Score computation methodology, particularly regarding differentiability challenges, and the specific effects of the hyperparameter `s` on generation quality and diversity.

**Strengths And Weaknesses:**

### Strengths

1. **Novel and effective:** The two-stage pipeline incorporating FRC, FVSG, and DSC is a valuable contribution. These methods appear to effectively tackle the distinct challenges of fairness, interclass diversity, and intraclass diversity.
2. **SOTA Performance:** VariFace demonstrates substantial improvements over existing synthetic datasets in constrained-size comparisons. Importantly, in unconstrained settings, it not only surpasses other synthetic methods but also outperforms models trained on the real CASIA-WebFace dataset, showcasing the power of synthetic data.
3. **Focus on fairness and diversity:** The paper explicitly addresses the critical issues of demographic bias and limited diversity in FR datasets. The method is designed to generate fair interclass variation and diverse intraclass variation. Figure 4 provides qualitative evidence of improved demographic representation compared to CASIA-WebFace and other synthetic datasets.
4. **Comprehensive Experimental Validation:** The authors conduct extensive experiments, including comparisons on standard benchmarks (LFW, CFP-FP, CPLFW, AgeDB, CALFW) and the RFW dataset for race-specific evaluation.

### Weaknesses

1. **Reliance on web-scraped data for training:** While the goal is to replace web-scraped training data for FR models, the VariFace generator itself is trained on CASIA-WebFace, and the authors acknowledge this limitation. While this is an improvement over directly training FR models on such data, the ethical concerns are not entirely eliminated but rather shifted.
2. **Definition and evaluation of "fairness":** Fairness in the paper is primarily defined as balanced demographic distributions for race and gender during Stage 1 sampling. While Table 2 results demonstrate improved performance for minority race categories, a deeper analysis of fairness, potentially including intersectional groups or other bias metrics beyond accuracy, could strengthen this aspect.
3. **Divergence score:** While DSC is an interesting way to control intraclass diversity without explicit attribute modeling, the "prototypical example" might itself be biased if the initial image set for an ID in the training data has skewed variations. How this might affect the learned diversity could be discussed.
4. **Qualitatives of intraclass variations:** Figure 3 shows DSC's effect, and Figure S5 shows age conditioning. More diverse qualitative examples showcasing the range of variations (pose, expression, illumination, accessories if any) generated for the _same_ identity would be interesting to display.
5. **Vendi Score:** The use of Vendi Score to increase diversity presents some challenges that require further discussion. First, the differentiability of Vendi Score is unclear, as it requires Singular Value Decomposition for eigenvalue computation, which is not generally differentiable. Second, Vendi Score measures ungrounded diversity and can be maximized by generating extremely different or out-of-domain samples, potentially compromising sample quality. The authors should clarify how these issues are addressed and explain why Vendi Score guidance appears to hurt quality results as shown in Table S1.
6. **Generation time:** There is a discrepancy in the reported generation times for VariFace between Section 4.6 (15 hours) and Table 5 (12 hours).

---

> ### Author Response · Authors · 2025-06-06
> **Reply to comments from Reviewer r5au**
>
> (Due to 5000 character limit, we will split our response into multiple comments).
>
> We are grateful for the reviewer’s thorough and valuable feedback. We are pleased to hear that the reviewer recognizes contributions of our method. We have updated the manuscript based on the feedback and here and we would like to address each concern and offer further clarifications:
>
> **Reliance on web-scraped data for training: While the goal is to replace web-scraped training data for FR models, the VariFace generator itself is trained on CASIA-WebFace, and the authors acknowledge this limitation. While this is an improvement over directly training FR models on such data, the ethical concerns are not entirely eliminated but rather shifted.** \
> While VariFace is trained on CASIA-WebFace, we have done this out of a necessity to provide a fair comparison against previous methods. CASIA-WebFace is one of the few remaining large-scale face datasets that has not been retracted and has therefore been used as the training data especially in recent synthetic face generation research. VariFace helps mitigate the privacy risks by avoiding training FR models directly on these web-scraped datasets, and we show empirically that most synthetic faces appear distinct from real faces (Figure S6). Currently, there is a lack of clean dataset alternatives that can achieve similar performances, but we believe that VariFace helps to bridge the gap towards developing fully synthetic datasets without the reliance on these web-scraped datasets.
>
> **Definition and evaluation of "fairness": Fairness in the paper is primarily defined as balanced demographic distributions for race and gender during Stage 1 sampling. While Table 2 results demonstrate improved performance for minority race categories, a deeper analysis of fairness, potentially including intersectional groups or other bias metrics beyond accuracy, could strengthen this aspect.** \
> We thank the reviewer for suggesting further fairness analysis. In response, we have extended our fairness evaluation beyond focusing on race and gender labels, to evaluate the skin color bias (Supplementary K). A limitation with the RFW race labels is that they are not comprehensive and do not account for the diversity within, and overlap between, each race category. In contrast, skin color, while not significant as race in terms of legal importance, provides a continuous and comprehensive scale to measure faces variation. Therefore, we believe skin color provides a complementary perspective to race and gender labels for evaluating the fairness of real and synthetic face datasets. Recently, a multidimensional scale [1] was proposed to analyze skin color bias, and evaluation of FFHQ and CelebAMask-HQ demonstrated that real face datasets display skin color bias and this is further exacerbated by synthetic datasets. We use their method to compute the skin color distribution of CASIA-WebFace, DCFace and VariFace (Results in Table S8). Our results highlight that while DCFace demonstrates similar skin color bias to CASIA-WebFace, VariFace not only reduces light-red skin color over-representation ($44.13\rightarrow33.13$) but also improves the minority dark-yellow skin color representation ($10.12\rightarrow14.39$). This demonstrates that VariFace not only reduces the bias with respect race labels, but generates a more diverse underlying distribution of individuals, including those reflecting underrepresented skin colors.
>
> **References** \
> [1] Thong, William, Przemyslaw Joniak, and Alice Xiang. "Beyond skin tone: A multidimensional measure of apparent skin color." Proceedings of the IEEE/CVF International Conference on Computer Vision. 2023.

---

> > ### Author Response · Authors · 2025-06-06
> > **Additional reply to comments from Reviewer r5au**
> >
> > **Divergence score: While DSC is an interesting way to control intraclass diversity without explicit attribute modeling, the "prototypical example" might itself be biased if the initial image set for an ID in the training data has skewed variations. How this might affect the learned diversity could be discussed.** \
> > Indeed, we find the skewed variation as an issue with CASIA-WebFace, but we believe that one of the reasons why DSC is effective is precisely through its ability to reduce this skewed variation. In real face datasets, some identities inevitably have little variation. For example, there will be no age variation for an identity if the images were all taken at the same time. In contrast, other identities may have more variation with images taken at different times, angles and locations. One of the key benefits of using DSC is that by implicitly learning intraclass variation over the dataset, when creating synthetic data, we can ensure consistent intraclass diversity across identities. This, however, also highlights a limitation of our method, which by consequence of its data-driven nature, means that if the original dataset does not have significant variation, our method will not be able to create diverse synthetic data. We have added this description to the limitations (Supplementary section L)
> >
> > **Qualitatives of intraclass variations: Figure 3 shows DSC's effect, and Figure S5 shows age conditioning. More diverse qualitative examples showcasing the range of variations (pose, expression, illumination, accessories if any) generated for the same identity would be interesting to display.** \
> > We thank the reviewer for the suggestion. We have added some further qualitative examples in the supplementary (Supplementary section M). These are unfiltered examples that demonstrate the capability of VariFace to generate a range of variation including, but not limited to, pose, expression, illumination and accessories (hats/glasses).
> >
> > **Vendi Score: The use of Vendi Score to increase diversity presents some challenges that require further discussion. First, the differentiability of Vendi Score is unclear, as it requires Singular Value Decomposition for eigenvalue computation, which is not generally differentiable. Second, Vendi Score measures ungrounded diversity and can be maximized by generating extremely different or out-of-domain samples, potentially compromising sample quality. The authors should clarify how these issues are addressed and explain why Vendi Score guidance appears to hurt quality results as shown in Table S1.** \
> > First, we would like to address the differentiability of the Vendi Score. The Vendi Score computation avoids the need to use SVD because we can make use of the fact that K is a symmetric, positive semi-definite matrix and retain full differentiability of the Vendi Score. Secondly, the reviewer is correct that generating out-of-domain samples would constitute an unfortunate result of maximizing the Vendi Score. To avoid this, the guidance scale parameter has to be carefully chosen, which balances the conditional denoising objective with the Vendi Score objective (Algorithm 1). This is somewhat analogous to the guidance scale used in Classifier-Free Guidance, where setting the guidance scale too high results in image quality degradation.
> >
> > **Generation time: There is a discrepancy in the reported generation times for VariFace between Section 4.6 (15 hours) and Table 5 (12 hours).** \
> > We apologize for the confusion. The 15 hours in section 4.6 refers to “the entire pipeline” i.e. both stage 1 and stage 2 processes. In Table 5, we compare the only “second stage generative model” inference times against DCFace and Vec2Face. The reason we benchmark the second stage inference times is because this is the bottleneck for the generative process. This is because while the first stage only generates a single image per identity, the second involves generating multiple (20-50) images per identity and constitutes the final synthetic dataset.

---

### Decision · Action_Editor_YEeT · 2025-08-16

**Recommendation:** Reject

**Additional Comments:**

The paper tackles an important problem with technically sound methods (FRC, FVSG, DSC) . But critical flaws require fixing:
- **Ethical rigor:** The privacy narrative is inconsistent. Must clarify how using FR models trained on non-consensual data (CASIA) aligns with privacy goals.
- **Fairness depth:** Expand beyond race/gender balance (e.g., age/intersectionality) and add bias metrics beyond accuracy.
- **Novelty positioning:** Compare rigorously with SynthDistill (two-stage similarity), CemiFace (diversity metrics), and ID³. Address Vendi Score’s non-differentiability and "ungrounded diversity" risks.
- **Clarity fixes:** Explain computational costs (resolve Table 5 vs Section 4.6 discrepancy), define "initial estimates" in Sec 3.1, and justify two-stage necessity.
- **Data equity:** Re-run CASIA comparisons with matched data volumes (e.g., 20K×50 like DCFace) to validate "outperforms real data" claims.

**Audience:**

Yes

**Audience Explanation:**

Yes. TMLR’s audience would find this highly relevant because:
- Synthetic data for privacy-sensitive FR is a critical unsolved problem in ML/CV.
- The pipeline’s focus on fairness/diversity (race/gender balance) addresses urgent ethical concerns.
- Demonstrated SOTA performance on FR benchmarks makes it practically impactful for real-world applications.
*But* ethical caveats (privacy loopholes, bias inheritance) need clearer disclosure to fully engage the community.

**Claims And Evidence:**

No

**Claims Explanation:**

Not completely yes with significant caveats. The claims about SOTA performance are supported by experiments across multiple benchmarks (LFW, CFP-FP, etc.), and qualitative results do show improved diversity/fairness (Fig 4). However:
- The claim *"outperforms real datasets"* is misleading – VariFace used 5× more data than CASIA-WebFace in "unconstrained" comparisons, skewing results.
- Only surface-level demographic balance was verified (race/gender), ignoring intersectionality or bias metrics beyond accuracy.
- Heavy reliance on pretrained FR models (derived from non-consensual web data) *in both stages* undermines the core privacy argument.
-  Key components (e.g., Divergence Score) overlap with missing references (CemiFace, ID³), and Vendi Score’s differentiability/grounding issues remain unaddressed.

**Resubmission Of Major Revision:**

The authors may consider submitting a major revision at a later time.